# BEST-OF-MAJORITY: MINIMAX-OPTIMAL STRATEGY FOR PASS@$k$ INFERENCE SCALING

**Qiwei Di**[*], **Kaixuan Ji**[*], **Xuheng Li**[*], **Heyang Zhao, Quanquan Gu**
Department of Computer Science
University of California, Los Angeles
CA 90095, USA
{qiwei2000, kaixuanji, xuheng.li, hyzhao, qgu}@cs.ucla.edu

## ABSTRACT

LLM inference often generates a batch of candidates for a prompt and selects one via strategies like majority voting or Best-of-$N$ (BoN). For difficult tasks, this single-shot selection often underperforms. Consequently, evaluations commonly report Pass@$k$: the agent may submit up to $k$ responses, and only the best of them is used when computing regret. Motivated by this, we study inference scaling in the more general Pass@$k$ inference setting, and prove that neither majority voting nor BoN exhibits the desirable scaling with $k$ and the sampling budget $N$. Combining the advantages of majority voting and BoN, we propose a new inference strategy called Best-of-Majority (BoM), with a pivotal step that restricts the candidates to the responses with high frequency in the $N$ samples before selecting the top-$k$ rewards. We prove that when the sampling budget is $N = \widetilde{\Omega}(C^*)$, the regret of BoM is $O(\epsilon_{\mathrm{opt}} + \sqrt{\epsilon_{\mathrm{RM}}^2 C^*/k})$, where $C^*$ is the coverage coefficient, $\epsilon_{\mathrm{RM}}$ is the estimation error of the reward model, and $\epsilon_{\mathrm{opt}}$ is the estimation error of reward at the optimal response. We further establish a matching lower bound, certifying that our algorithm is minimax optimal. Beyond optimality, BoM has a key advantage: unlike majority voting and BoN, its performance does not degrade when increasing $N$. Experimental results of inference on math problems show BoM outperforming both majority voting and BoN.

## 1 INTRODUCTION

Scaling law serves as a powerful tool for guiding the *training* of large language models (LLMs), providing insight into how increased training compute, data, and model size contribute to performance improvements. Originating in the early days of deep neural networks (Hestness et al., 2017; Rosenfeld et al., 2019), the concept has since demonstrated remarkable predictive power across a variety of domains, including strategic board games (Jones, 2021), image generation (Henighan et al., 2020; Yu et al., 2022; Peebles & Xie, 2023), video modeling (Brooks et al., 2024), language generation (Kaplan et al., 2020; Hoffmann et al., 2022; Achiam et al., 2023), retrieval systems (Fang et al., 2024; Cai et al., 2025), and reward modeling (Gao et al., 2023; Rafailov et al., 2024). While training-time scaling has proven effective, it is also highly resource-intensive. As a result, increasing attention has been directed toward a complementary paradigm: *inference*, which examines how model performance can be improved after training. This relationship between additional compute at inference time and performance improvement is known as the inference scaling law (Brown et al., 2024; Snell et al., 2024; Wu et al., 2024b; Guo et al., 2025).

Compared to training-time scaling, inference scaling allows for increasing computational cost in several distinct ways, including expanding the generation input via chain-of-thought prompting (Wei et al., 2022; Li et al., 2024), incorporating iterative self-improvement, (Zheng et al., 2023; Wu et al., 2024a), and applying search-based algorithms (Yao et al., 2023; Feng et al., 2023; Gao et al., 2024; Zhang et al., 2024). It can also be realized through repeated sampling, using strategies such as majority voting (Wang et al., 2022; Lewkowycz et al., 2022; Li et al., 2023) or Best-of-$N$ (BoN) (Lightman et al., 2023). In parallel, a growing line of works has sought to establish theoretical guarantees for inference strategies. Wu et al. (2024b) provided convergence bounds and rates for the scaling of majority voting algorithms. Huang et al. (2024) showed that BoN can achieve self-improvement via a special mechanism called sharpening. Huang et al. (2025a) analyzed the sample

---

[*]Equal contribution

Table 1: Comparison of Pass@$k$ inference strategies. Our algorithm BoM is the first minimax-optimal Pass@$k$ inference strategy. Compared with majority voting and BoN, BoM is scaling-monotonic, indicating that the optimal performance can be achieved with large sampling budget $N$, making it preferable when scaling up $N$ to achieve better performance. Additionally, the term $O(\sqrt{\epsilon_{\mathrm{RM}}^2 C^*/k})$ in the regret of BoM scales optimally with $k$, while majority voting suffers from constant regret. BoN lacks the regret upper bound in the Pass@$k$ inference problem.

| Algorithm | Worst-case regret | Scaling-monotonic | Optimal $k$-scaling |
|---|---|---|---|
| Majority voting | $\Omega(1)$ 
 (Theorem 4.1) | No | No |
| Best-of-$N$ | $\Omega(\min\{1, \sqrt{\epsilon_{\mathrm{RM}}^2 N/k}\})$ 
 (Theorem 4.2 | No | Unknown |
| Best-of-Majority (Ours) | $O(\epsilon_{\mathrm{opt}} + \sqrt{\epsilon_{\mathrm{RM}}^2 C^*/k})$ 
 (Theorem 5.1) | Yes | Yes |
| Lower Bound | $\Omega(\epsilon_{\mathrm{opt}} + \sqrt{\epsilon_{\mathrm{RM}}^2 C^*/k})$ 
 (Theorem 6.1) | - | - |

complexity of BoN and proposed a pessimistic inference algorithm with provable benefits. Chen et al. (2025a) bridged test-time computation with theoretical analysis of transformers in the setting of in-context learning of linear regression.

While most existing analyses focus on inference algorithms that output a single response, there are tasks that allow for multiple candidate outputs, where it is considered solved if any one of them is correct. This setting is captured by the Pass@$k$ metric (Li et al., 2022). Building on this metric, we propose a novel **Pass@$k$** inference framework, in which the inference algorithm is allowed to generate $N$ responses and return up to $k$ of them. Since $N > k$, the performance depends not only on generating a diverse set of candidates but also on the algorithm's ability to effectively select the $k$ outputs that are most likely to be correct. Brown et al. (2024) conducted empirical studies on this inference framework and observed the relationship between the coverage and the performance of the algorithm. However, this work is restricted to the majority voting and BoN inference strategies, and failed to theoretically justify the inference scaling law.

As there have been few works on understanding the scaling of the Pass@$k$ inference problem, we are motivated to investigate the following fundamental question:

*Q1: What is the optimal scaling of the Pass@$k$ inference problem?*

To answer this question, we derive a minimax lower bound as a function of $k$ that characterizes the fundamental limits of any Pass@$k$ inference strategy, establishing the theoretical scaling behavior for Pass@$k$ inference problems.

Going one step further, we also aim to evaluate existing inference strategies for the Pass@$k$ inference problem and find a strategy that achieves the optimal scaling. Beyond standard metrics like regret and sample complexity, we further introduce a formal definition of *scaling-monotonicity* (Huang et al., 2025a), which captures whether an inference algorithm maintains (or improves) its performance as the number of samples $N$ increases. This leads to our second question:

*Q2: What inference strategies are scaling-monotonic and optimal in the Pass@$k$ inference setting?*

Unfortunately, our analysis reveals that majority voting and BoN are not scaling-monotonic. Furthermore, these methods face fundamental limitations that make it difficult, if not impossible, to attain the optimal regret scaling with respect to $k$. To address this issue, we propose a new inference strategy, Best-of-Majority (BoM), which integrates the core ideas of both majority voting and BoN. We establish a regret upper bound for BoM that matches the minimax lower bound, thereby demonstrating that our algorithm is minimax optimal. Please refer to Table 1 for detailed results.

We summarize our main contributions as follows:

- **Inference scaling laws for Pass@$k$.** We show that the minimax lower bound of the regret is $\Omega(\epsilon_{\mathrm{opt}} + \sqrt{\epsilon_{\mathrm{RM}}^2 C^*/k})$ for any Pass@$k$ inference strategy, where $\epsilon_{\mathrm{opt}}$ is the error of the reward model at the optimal response, $\epsilon_{\mathrm{RM}}$ is the expected error of the reward model, and $C^*$ is the coverage of the reference LLM.

- **Optimal algorithm for Pass@$k$.** We propose a new Pass@$k$ inference strategy called Best-of-Majority (BoM). At the core of BoM is a step similar to majority voting that restricts the candidates

to the responses with high frequencies in the generated samples, before selecting responses with top-$k$ rewards. We prove that the regret of BoM is $O(\epsilon_{\text{opt}} + \sqrt{\epsilon_{\text{RM}}^2 C^*/k})$ with sample complexity $N = \widetilde{\Theta}(C^*)$, thus matching the minimax lower bound without increasing the computation overhead. With a formal definition of scaling monotonicity, we show that BoM is scaling monotonic, while majority voting and BoN are not.

- **Experiments.** We compare our algorithm BoM against majority voting and BoN. Our results empirically demonstrate the superiority of BoM against majority voting and BoN and verify the scaling monotonic properties of three algorithms, which corroborates our theoretical results.

**Notations.** We use $[M]$ to denote the set of integers $\{1, 2, \ldots, M\}$. We use $\mathbb{1}[\cdot]$ to denote the indicator function. We use $\delta_{ij}$ to denote the Kronecker delta, i.e., $\delta_{ij} = 1$ if $i = j$, and $\delta_{ij} = 0$ otherwise. We use $y, y_i$ to denote the elements in the set of response $\mathcal{Y}$, $\widehat{y}, \widehat{y}_i$ to denote the generated responses, and $\widetilde{y}, \widetilde{y}_i$ to denote the final outputs. We use standard asymptotic notations $O(\cdot), \Omega(\cdot)$, and $\Theta(\cdot)$, and use $\widetilde{O}(\cdot), \widetilde{\Omega}(\cdot)$ and $\widetilde{\Theta}(\cdot)$ to further hide the logarithmic factors.

## 2 RELATED WORK

**Inference-time scaling.** Compared to training-time scaling laws, the study of inference-time scaling laws has emerged much more recently. Sardana et al. (2024) extended the Chinchilla scaling law (Hoffmann et al., 2022) to incorporate inference costs. Wu et al. (2024b) conducted a systematic study of inference scaling laws, analyzing a range of inference strategies including greedy search, majority voting, best-of-$N$, weighted voting, and two variants of tree-based search algorithms. Concurrently, Snell et al. (2024) analyzed the inference scaling problem by searching against process-based verifier reward models. In contrast, Brown et al. (2024) explored repeated sampling as a simple scaling method to improve performance. Chen et al. (2024) studied the performance of majority voting and a variant that incorporates a filtering mechanism. They observed that as the number of generated samples $N$ increases, performance initially improves but eventually declines. They also proposed a predictive scaling model to characterize the performance trend. Muennighoff et al. (2025) developed simple methods to construct a sample-efficient test-time scaling dataset. Moreover, Huang et al. (2025b) studied the sample complexity and representation ability of test-time scaling with transformers.

**Inference strategies.** One of the most straightforward inference strategies is best-of-$N$, which has been widely adopted in the inference of language models (Stiennon et al., 2020; Nakano et al., 2021; Touvron et al., 2023; Gao et al., 2023). For its theoretical guarantees, Yang et al. (2024a) established a connection between the asymptotic behavior of BoN and KL-constrained reinforcement learning methods, characterizing this relationship through information-theoretic quantities. Beirami et al. (2024) provided a tighter upper bound for the KL divergence between the BoN policy and the reference policy. Mroueh (2024) proved guarantees for BoN algorithm from a information theoretic view. Huang et al. (2025a) further provided guarantees on performance when the estimated reward model and true reward are mismatched. Aminian et al. (2025) extended the analysis to a smoothed variant of BoN called SBoN. They proved that BoN can suffer from reward overoptimization, and SBoN can serve as a mitigation. Another common inference strategy is majority voting (Lewkowycz et al., 2022; Wang et al., 2022; Li et al., 2023). Wu et al. (2024b) established convergence bounds and rates characterizing how the performance of majority voting algorithms scales with the number of samples. Other inference strategies include variants of BoN (Jinnai et al., 2024; Qiu et al., 2024), rejection sampling (Liu et al., 2023; Xu et al., 2024), and search-based algorithms (Yao et al., 2023; Feng et al., 2023; Gao et al., 2024; Zhang et al., 2024).

**Pass@$k$ alignment.** To the best of our knowledge, the theoretical Pass@$k$ inference framework is novel and remains unexplored in the existing literature. Moreover, aligning the training process with different inference algorithms has also emerged as a promising direction (Balashankar et al., 2024). In this direction, Pass@$k$ has also been proved useful in the training of large language models. Tang et al. (2025) demonstrated that training language models using a Pass@$k$-based objective can lead to improved overall model performance. More recently, Chen et al. (2025b) used Pass@$k$ as the reward to train the language model and observe improvements on its exploration ability. Liang et al. (2025) proposed training methods to mitigate entropy collapse, which in turn lead to improved performance on the Pass@$k$ metric.

## 3 PASS@$k$ INFERENCE SCALING PROBLEM

Let $\mathcal{X}$ be the set of prompts and $\mathcal{Y}$ the set of responses. We represent an LLM as a conditional policy $\pi(\cdot \mid x)$ that maps each prompt $x \in \mathcal{X}$ to a distribution over $\mathcal{Y}$. We have access to a reference policy $\pi_{\text{ref}}$, which, for instance, can be trained using the supervised finetuning (SFT) method. For each pair $(x, y) \in \mathcal{X} \times \mathcal{Y}$, we assume the existence of a ground-truth reward model $r^* : \mathcal{X} \times \mathcal{Y} \to [0, 1]$, which evaluates the quality of response $y$ given prompt $x$.

During inference time, we can use the reference policy $\pi_{\text{ref}}$ to generate multiple responses. To evaluate the quality of these responses, we utilize an imperfect reward model $\widehat{r} : \mathcal{X} \times \mathcal{Y} \to [0, 1]$, which provides approximate assessments of response quality. For a given prompt $x$, we make the following assumptions regarding the accuracy of the reward model.

**Assumption 3.1** (Reward Estimation Error). The expected squared error between $r^*$ and $\widehat{r}$ is upper bounded by $\epsilon_{\text{RM}}^2(x)$, i.e,

$$\mathbb{E}_{y \sim \pi_{\text{ref}}(\cdot|x)}\left[\left(r^*(x, y) - \widehat{r}(x, y)\right)^2\right] \leq \epsilon_{\text{RM}}^2(x).$$

This assumption is the standard squared loss of the reward model, with respect to the reference policy $\pi_{\text{ref}}$. The same assumption has been made in prior work like Huang et al. (2025a).

We further assume that the optimal answer is unique, which is a natural condition in many domains where the correctness of the final solution is verifiable, such as in mathematical problems.

**Assumption 3.2.** There exists a unique $y^* = \text{argmax}_{y \in \mathcal{Y}} r^*(x, y)$, with $r^*(x, y^*) = 1$. Moreover, the estimated reward at $y^*$ is close to optimal, satisfying

$$|r^*(x, y^*) - \widehat{r}(x, y^*)| = \epsilon_{\text{opt}}(x).$$

Combining Assumption 3.1 with Assumption 3.2, we directly know $\pi_{\text{ref}}(y^*|x) \cdot \epsilon_{\text{opt}}^2(x) \leq \epsilon_{\text{RM}}^2(x)$.

In practice, an accurate reward model is crucial for the post-training and inference of large language models. A common approach is to align the model with human preference data through supervised learning or reinforcement learning from human feedback (RLHF) (Ouyang et al., 2022; Casper et al., 2023; Zhu et al., 2024; Yang et al., 2024b). Since the training of the reward model extensively studied and is not the focus of this work, we directly assume access to a pre-training reward model that satisfies Assumptions 3.1 and 3.2.

In this work, we study a novel setting called the **Pass@$k$** inference scaling problem. Different from the settings where the model is allowed to generate and submit $k$ candidate responses, our goal is to maximize the highest ground-truth reward of the $k$ samples. Specifically, for a given prompt $x$, the model is allowed to generate up to $N$ candidate responses and select a subset $y_1, y_2, \ldots, y_k$ for submission. Increasing $N$ improves the likelihood of obtaining high-quality outputs, but also incurs greater computational cost, a trade-off between accuracy and efficiency. We consider the following regret metric:

$$\text{Regret}(x) = \mathbb{E}_{\pi^*}\left[r^*(x, \cdot)\right] - \mathbb{E}_{y_1, y_2, \ldots, y_k}\left[\max_{1 \leq i \leq k} \{r^*(x, y_i)\}\right], \tag{3.1}$$

where $\pi^* = \pi^*(\cdot|x)$ is the maximizer of $r^*$.

In tasks with a unique correct answer, such as mathematical problem solving, the ground-truth reward model $r^*$ functions as a binary verifier, returning values in $\{0, 1\}$. In this case, the regret (3.1) naturally aligns with the Pass@$k$ metric (Li et al., 2022), since minimizing (3.1) is equivalent to maximizing the probability that at least one of the $k$ selected responses is correct.

**Remark 3.3.** Compared with the sample-and-evaluate framework (Huang et al., 2025a), our framework goes one step further by explicitly characterizing the dependence on $k$. This dependence constitutes a novel focus of our analysis, as it has not been examined in prior works on inference-time algorithms (Huang et al., 2024; 2025a; Verdun et al., 2025).

In the setting of test-time (inference) scaling, responses are generated by $\pi_{\text{ref}}$. Sampling from a good reference model can lead to performance improvement. Motivated by this, we need a metric to evaluate the performance of the reference model. Following Huang et al. (2025a), we introduce the reference policy's $L_1$-coverage coefficient as follows:

$$C^*(x) := \mathbb{E}_{y \sim \pi^*(\cdot|x)}\left[\pi^*(y|x)/\pi_{\text{ref}}(y|x)\right]. \tag{3.2}$$

Moreover, the uniform coverage coefficient is defined as

$$C_\infty^*(x) := \sup_y \left[ \pi^*(y|x)/\pi_{\text{ref}}(y|x) \right]. \tag{3.3}$$

Since Assumption 3.2 ensures that the optimal policy $\pi^*$ is deterministic and uniquely defined as $\pi^*(y|x) = \mathbb{1}(y = y^*)$, the $L_1$ and uniform coverage coefficients coincide. Consequently, we have $C^*(x) = C_\infty^*(x) = 1/\pi_{\text{ref}}(y^*|x)$.

Besides the regret, we are also concerned with the following important property of the algorithm, named as *scaling-monotonicity* (Huang et al., 2025a). We provide the formal definition as follows:

**Definition 3.4.** Assume that $k$, prompt $x$ and the coverage coefficient $C^*(x)$ are fixed. An algorithm is *scaling-monotonic* if for any $\delta > 0$, there exists $\epsilon_0 > 0$ and $N_0 \in \mathbb{N}_+$ such that for any $N \geq N_0$ and any instance that satisfies Assumption 3.1 with $\epsilon_{\text{RM}}(x) \leq \epsilon_0$, the regret satisfies

$$\text{Regret}(x) \leq \delta.$$

Intuitively, a scaling-monotonic algorithm should achieve arbitrarily small regret if the reward model $\widehat{r}$ is accurate and sufficiently many samples of responses from $\pi_{\text{ref}}$ are observed. Furthermore, scaling monotonicity also guarantees that the performance of the algorithm does not degrade when increasing $N$. Therefore, it is a crucial property in practice because the sampling budget $N$ can be easily scaled up in hard instances instead of requiring accurate tuning. While this concept is not completely new, as far as we know, we are the first to formally define this property.

## 4 SUBOPTIMALITY OF EXISTING INFERENCE STRATEGIES

In this section, we first introduce two commonly used strategies for LLM inference, namely (weighted) majority voting (Section 4.1) and Best-of-$N$ (BoN, Section 4.2). We will show that neither strategy is scaling-monotonic by constructing hard instances where the inference strategies suffer from constant regret even when $N \to \infty$. Additionally, the Pass@$k$ inference problem is less stringent than Pass@1, since it only requires success in any of the $k$ sampled attempts rather than a single one. Consequently, the regret is expected to decrease as $k$ increases, suggesting a negative association between regret and the sampling budget $k$.

### 4.1 (WEIGHTED) MAJORITY VOTING

Majority voting is a simple ensemble method for LLM inference: Multiple responses to the same prompt are sampled using the reference policy $\pi_{\text{ref}}(\cdot|x)$ to make the responses diverse enough, and the answer occurring most often is selected as the final output.

Specifically, let $\widehat{y}_1, \ldots, \widehat{y}_N$ denote the $N$ generated responses for a given query. After calculating the frequency of each response $\widehat{\pi}(y) = \frac{1}{N} \sum_{i=1}^N \mathbb{1}(\widehat{y}_i = y)$, the final prediction is then chosen as the answer that appears most frequently among these samples, i.e.,

$$\widetilde{y}_1, \ldots, \widetilde{y}_k = \text{Top-k}\{y \in \widehat{\mathcal{Y}} : \widehat{\pi}(y)\}.$$

Majority voting has demonstrated strong empirical performance (Wang et al., 2022; Lewkowycz

---

**Algorithm 1** (Weighted) Majority Voting

**Require:** Reference policy $\pi_{\text{ref}}$, sampling budget $N$, number of candidates $k$, (estimated reward model $\widehat{r}$, weight function $w(\cdot)$).
1: Observe context $x$.
2: Independently generate $N$ responses $\widehat{\mathcal{Y}} = \{\widehat{y}_1, \widehat{y}_2, \ldots, \widehat{y}_N\}$ from $\pi_{\text{ref}}(\cdot|x)$.
3: **if** $|\widehat{\mathcal{Y}}| \leq k$ **then**
4:     **return** $\widehat{\mathcal{Y}}$.
5: **else**
6:     Calculate frequency of each response $y \in \widehat{\mathcal{Y}}$: $\widehat{\pi}(y) = \frac{1}{N} \sum_{i=1}^N \mathbb{1}[\widehat{y}_i = y]$.
7:     **if** *weighted* **then**
8:         Query reward labels $(\widehat{r}(x, \widehat{y}_1), \ldots, \widehat{r}(x, \widehat{y}_N))$
9:         Select $\widetilde{y}_1, \ldots, \widetilde{y}_k = \text{Top-k}\{y \in \widehat{\mathcal{Y}} : w(\widehat{r}(y)) \cdot \widehat{\pi}(y)\}$.
10:    **else**
11:        Select $\widetilde{y}_1, \ldots, \widetilde{y}_k = \text{Top-k}\{y \in \widehat{\mathcal{Y}} : \widehat{\pi}(y)\}$.
12:    **end if**
13:    **return** $\{\widetilde{y}_1, \ldots, \widetilde{y}_k\}$.
14: **end if**

---

et al., 2022; Li et al., 2023). With a reliable reward model $\widehat{r}$, it can be further enhanced by weighting candidate frequencies with reward scores. Using an increasing weighting function $w(\cdot)$, the selection rule becomes:

$$\widetilde{y}_1, \ldots, \widetilde{y}_k = \text{Top-k}\{y \in \widehat{\mathcal{Y}} : w(\widehat{r}(y)) \cdot \widehat{\pi}(y)\}.$$

While the reward weighting introduces extra computation for reward evaluation, weighted majority voting has been shown to achieve better performance than the unweighted version (Wu et al., 2024b). Despite its empirical success, we show that (weighted) majority voting is suboptimal in the worst case, even when the exact reward function is available, i.e., $\epsilon_{\text{RM}}^2(x) = 0$.

**Theorem 4.1.** For the (weighted) majority voting Algorithm 1 with weight function $w(\cdot)$, assume that $C^*(x) \geq 1 + 2kw(1)/w(1/2)$. Then, there exists an instance $\mathcal{I} = (\mathcal{X}, \mathcal{Y}, \pi^*, r^*, \pi_{\text{ref}}, \widehat{r})$ such that the coverage coefficient is $C^*(x)$, and $\widehat{r} = r^*$ satisfies Assumptions 3.1 and 3.2 with $\epsilon_{\text{RM}}(x) = \epsilon_{\text{opt}}(x) = 0$. If $N \geq 9C^*(x)\log(2k+2)$, the algorithm suffers from a constant regret:

$$\text{Regret}(x) = \Omega(1).$$

Majority voting relies on exploiting the reference model's distribution. Consequently, the hard case can be constructed by designing multiple distinct "bad" answers, each receiving higher probability under $\pi_{\text{ref}}$ than the probability of the optimal answer $\pi_{\text{ref}}(y^*)$. Theorem 4.1 demonstrates that increasing the sampling budget $N$ or the number of submitted responses $k$ does not guarantee consistent improvement for (weighted) majority voting. In fact, when $N$ is sufficiently large, (weighted) majority voting incurs constant regret even if the reward model is accurate.

### 4.2 BEST-OF-$N$

Best-of-$N$ is another effective LLM inference strategy. Instead of aggregating answers by frequency, the model generates multiple candidate responses for the same query and then selects the single best response according to a reward model $\widehat{r}$. Formally, given $N$ sampled responses $\widehat{y}_1, \ldots, \widehat{y}_N$, the Best-of-$N$ strategy selects the outputs that maximize the reward signal $\widehat{r}$, i.e.,

$$\widetilde{y}_1, \ldots, \widetilde{y}_k = \text{Top-k}\big\{y \in \widehat{\mathcal{Y}} : \widehat{r}(y)\big\}.$$

For the BoN algorithm, we have the following theorem on the lower bound of the regret.

---

**Algorithm 2** Best-of-$N$ (BoN)

**Require:** Estimated reward model $\widehat{r}$, reference policy $\pi_{\text{ref}}$, sampling budget $N$, number of candidates $k$.
1: Observe context $x$.
2: Independently generate $N$ responses $\widehat{\mathcal{Y}} = \{\widehat{y}_1, \widehat{y}_2, \ldots, \widehat{y}_N\}$ from $\pi_{\text{ref}}(\cdot|x)$.
3: Query reward labels $(\widehat{r}(x, y_1), \ldots, \widehat{r}(x, y_N))$.
4: **if** $|\widehat{\mathcal{Y}}| \leq k$ **then**
5:     **return** $\widehat{\mathcal{Y}}$.
6: **else**
7:     Select $\widetilde{y}_1, \ldots, \widetilde{y}_k = \text{Top-k}\big\{y \in \widehat{\mathcal{Y}} : \widehat{r}(x, y)\big\}$.
8:     **return** $\{\widetilde{y}_1, \ldots, \widetilde{y}_k\}$.
9: **end if**

---

**Theorem 4.2.** For BoN (Algorithm 2), assume that $C^*(x) \geq 2k$. Then, there exists an instance $\mathcal{I} = (\mathcal{X}, \mathcal{Y}, \pi^*, r^*, \pi_{\text{ref}}, \widehat{r})$ such that the coverage coefficient is $C^*(x)$, and $(\widehat{r}, r^*)$ satisfies Assumptions 3.1 and 3.2 with $\epsilon_{\text{RM}}(x)$ and $\epsilon_{\text{opt}}(x)$. If $N \leq C^*(x)$, Algorithm 2 suffer from a constant regret, i.e.,

$$\text{Regret}(x) = \Omega(1).$$

Otherwise, the regret satisfies

$$\text{Regret}(x) = \Omega\Big(\min\Big\{1, \sqrt{N\epsilon_{\text{RM}}^2(x)/k}\Big\}\Big).$$

BoN leverages the reward model's signal, but this makes it vulnerable to reward overoptimization (Gao et al., 2023; Stroebl et al., 2024) when the reward model is inaccurate. Thus, we construct the hard case by introducing multiple distinct "bad" answers that are assigned higher estimated rewards. With a carefully chosen, problem-dependent sampling budget $N = \widetilde{\Theta}(C^*(x))$, the lower bound will become $\widetilde{\Omega}(\sqrt{C^*(x)\epsilon_{\text{RM}}^2(x)/k})$, which aligns with the general lower bound for inference algorithms (as will be discussed in Section 6). However, this lower bound implies that BoN is not scaling-monotonic, as for fixed $k$ and $\epsilon_{\text{RM}}(x)$, the regret converges to a non-zero constant when $N$ becomes sufficiently large. Thus, increasing $N$ for BoN not only causes higher computational overhead, but can also degrade performance when the reward model is inaccurate.

**Remark 4.3.** For the case where $k = 1$, we establish a lower bound of $\Omega(\min\{1, \sqrt{N\epsilon_{\text{RM}}^2(x)}\})$, which recovers Theorem 3.2 in Huang et al. (2025a). Notably, Huang et al. (2025a) also provided a matching upper bound when $k = 1$. However, for $k > 1$, our analysis reveals an additional factor of $1/\sqrt{k}$ in the lower bound, which has not been considered in prior works. A direct application of the upper bound analysis from Huang et al. (2025a) fails to capture this dependence on $k$, leaving a theoretical gap of $1/\sqrt{k}$. It remains unclear whether this suboptimality is due to limitations in the analysis or is an inherent weakness of BoN. We conjecture that obtaining a regret upper bound for BoN with the optimal $1/\sqrt{k}$ scaling under the Pass@$k$ setting may be fundamentally infeasible. We leave this to future work.

## 5 OPTIMAL ALGORITHM FOR PASS@K INFERENCE

In Section 4, we have proved that neither (weighted) majority voting nor BoN is scaling monotonic, and neither demonstrates the desirable scaling with $k$ for the Pass@$k$ inference scaling problem. We also explicitly describe how the corresponding hard instance is constructed. By contrast, instances that do not satisfy the properties of these hard cases are precisely those on which the algorithms perform well. Thus, our earlier analysis also reveals complementary strengths of these methods: majority voting performs well when the reference policy assigns a higher probability to the ground-truth answer than to incorrect ones, while Best-of-$N$ can be highly effective when the reward model $\widehat{r}$ is accurate. However, each method also exhibits weaknesses, as they fail to fully exploit the available information from either the policy or the reward model. To address these limitations, we introduce a new algorithm, Best-of-Majority (BoM), which integrates the advantages of both approaches.

Our algorithm is built upon the principles of pessimism commonly used in reinforcement learning (Buckman et al., 2020; Jin et al., 2021). When the reference policy $\pi_{\text{ref}}$ assigns low probability to a response, that response is rarely observed in the training data. Consequently, the reward model receives limited supervision in this region, leading to higher uncertainty and likelihood of error. The pessimism principle advocates making conservative predictions under such uncertainty, which motivates our design choice: we rely on the reward model only when $\pi_{\text{ref}}$ assigns sufficiently high probability to the candidate. Since $\pi_{\text{ref}}$ cannot be directly observed, we approximate it using empirical frequencies of generated responses. Specifically, let $\widehat{y}_1, \ldots, \widehat{y}_N$ denote the $N$ generated responses for a given query. We first calculate the empirical frequency of each emerging response:

---

**Algorithm 3** Best-of-Majority (BoM)

**Require:** Estimated reward model $\widehat{r}$, reference policy $\pi_{\text{ref}}$, frequency threshold $\alpha$, sampling budget $N$, number of candidates $k$.

1: Observe context $x$.
2: Independently generate $N$ responses $\widehat{\mathcal{Y}} = \{\widehat{y}_1, \widehat{y}_2, \ldots, \widehat{y}_N\}$ from $\pi_{\text{ref}}(\cdot|x)$.
3: Calculate frequency of each response $y \in \mathcal{Y}$: $\widehat{\pi}(y) = \frac{1}{N}\sum_{i=1}^{N} \mathbb{1}(\widehat{y}_i = y)$.
4: Eliminate responses with frequency less than $\alpha$: $\widehat{\mathcal{Y}}_\alpha = \{y \in \widehat{\mathcal{Y}} : \widehat{\pi}(y) \geq \alpha\}$.
5: Query reward labels $(\widehat{r}(x, \widehat{y}_1), \ldots, \widehat{r}(x, \widehat{y}_N))$.
6: **if** $|\widehat{\mathcal{Y}}_\alpha| \leq k$ **then**
7:     **return** $\widehat{\mathcal{Y}}_\alpha$.
8: **else**
9:     Select $\widetilde{y}_1, \ldots, \widetilde{y}_k = \text{Top-k}\{y \in \widehat{\mathcal{Y}}_\alpha : \widehat{r}(y)\}$.
10:     **return** $\{\widetilde{y}_1, \ldots, \widetilde{y}_k\}$.
11: **end if**

---

$$\widehat{\pi}(y) = \frac{1}{N}\sum_{i=1}^{N} \mathbb{1}(\widehat{y}_i = y).$$

Guided by the pessimism principle, we discard responses whose frequency falls below a threshold $\alpha$, retaining only the subset

$$\widehat{\mathcal{Y}}_\alpha = \{y \in \widehat{\mathcal{Y}} : \widehat{\pi}(y) \geq \alpha\}.$$

Then we query the reward model on the surviving candidates and select the top $k$ responses according to their predicted rewards, $\widetilde{y}_1, \ldots, \widetilde{y}_k = \text{Top-k}\{y \in \widehat{\mathcal{Y}}_\alpha : \widehat{r}(y)\}$. The following theorem demonstrates the upper bound of BoM.

**Theorem 5.1.** Assume that the threshold is $\alpha = 3/(4C^*(x))$. Then the regret of BoM (Algorithm 3) satisfies

$$\text{Regret}(x) \leq \epsilon_{\text{opt}}(x) + O\left(\sqrt{C^*(x)\epsilon_{\text{RM}}^2(x)/k}\right) + O\left(C^*(x)e^{-N/(32C^*(x))}\right).$$

Moreover, when the sampling budget satisfies $N \geq 16C^*(x)\log\left(kC^*(x)/\epsilon_{\text{RM}}^2(x)\right)$, we have

$$\text{Regret}(x) \leq \epsilon_{\text{opt}}(x) + \widetilde{O}\left(\sqrt{C^*(x)\epsilon_{\text{RM}}^2(x)/k}\right).$$

Here, we let $N \geq 16C^*(x)\log\left(kC^*(x)/\epsilon_{\text{RM}}^2(x)\right)$ with a term dependent on $\epsilon_{\text{RM}}(x)$ to simplify the results. In general, we can replace the $\epsilon_{\text{RM}}(x)$ with any accuracy $\epsilon$ that we wish to achieve. Thus, Our selection of $N$ do not require the knowledge of $\epsilon_{\text{RM}}(x)$, and our results will continue to work when $\epsilon_{\text{RM}}(x) = 0$. When $\epsilon_{\text{opt}}(x) \ll \sqrt{C^*(x)\epsilon_{\text{RM}}^2(x)}$, the second term dominates, and consequently the overall regret scales as $1/\sqrt{k}$, consistent with the intuition that increasing $k$ enlarges the candidate set and thereby makes the problem easier. Moreover, for fixed $x$, $k$, and $C^*(x)$, the regret bound converges to 0 as $N \to \infty$ and $\epsilon_{\text{RM}}(x) \to 0$. This yields the following corollary.

**Corollary 5.2.** BoM (Algorithm 3) is scaling-monotonic.

**Computational Complexity.** According to Theorem 5.1, the BoM algorithm requires approximately $\widetilde{\Omega}(C^*(x))$ samples to achieve low regret. In comparison, Theorem 3.4 in Huang et al. (2025) shows that when $k = 1$, the Best-of-$N$ (BoN) algorithm also requires $\widetilde{\Theta}(C^*(x))$ samples. This means for Pass@$k$ inference, BoM achieves a better regret bound with a $1/\sqrt{k}$ improvement without incurring additional generation cost. Moreover, BoM only queries the reward model for a filtered subset of candidates (see Algorithm 3, Line 5), which can reduce the number of reward evaluations.

*Proof Sketch of Theorem 5.1.* The crucial step of BoM involves the construction of $\widehat{\mathcal{Y}}_\alpha$ to approximate the set of all responses $y$ with $\pi_{\text{ref}}(y|x) \geq \alpha$, denoted by $\mathcal{Y}_\alpha$. The following two properties of $\mathcal{Y}_\alpha$ makes it preferable as the set of candidates: Firstly, if $\widetilde{y}_i \in \mathcal{Y}_\alpha(x)$ for all $i \in [k]$, we have an upper bound of the minimum estimation error $\min_{i \in [k]} \Delta_i$, where $\Delta_i = |\widehat{r}(x, \widetilde{y}_i) - r^*(x, \widetilde{y}_i)|$:

$$\min_{i \in [k]} \Delta_i \leq \sqrt{\sum_{i=1}^k \Delta_i^2/k} \leq \sqrt{\sum_{i=1}^k \pi_{\text{ref}}(\widetilde{y}_i|x)\Delta_i^2/(\alpha k)} \leq \sqrt{\epsilon_{\text{RM}}^2(x)/(\alpha k)}, \quad (5.1)$$

where we used the property $\pi_{\text{ref}}(\widetilde{y}_i|x) \geq \alpha$ in the second inequality. Secondly, since $\pi_{\text{ref}}(y^*|x) \geq 1/C^*(x)$, we have $y^* \in \mathcal{Y}_{1/C^*(x)}$. Therefore, if $\widehat{\mathcal{Y}}_\alpha(x) = \mathcal{Y}_{1/C^*(x)}(x)$, the algorithm either outputs $y^*$ among the $k$ submitted responses, incurring zero regret, or outputs $k$ responses with $\widehat{r}(x, \widetilde{y}_i) \geq \widehat{r}(x, y^*)$, where the regret can be decomposed as

$$r^*(x, y^*) - r^*(x, \widetilde{y}_i) \leq \underbrace{|r^*(x, y^*) - \widehat{r}(x, y^*)|}_{\epsilon_{\text{opt}}(x)} + \underbrace{[\widehat{r}(x, y^*) - \widehat{r}(x, \widetilde{y}_i)]}_{\leq 0} + \underbrace{|\widehat{r}(x, \widetilde{y}_i) - r^*(x, \widetilde{y}_i)|}_{\Delta_i}.$$

We take the minimum, plug in (5.1), and obtain

$$r^*(x, y^*) - \max_{i \in [k]} r^*(x, \widetilde{y}_i) \leq \epsilon_{\text{opt}}(x) + \min_{i \in [k]} \Delta_{\widetilde{y}_i} \leq \epsilon_{\text{opt}}(x) + \sqrt{4C^*(x)\epsilon_{\text{RM}}^2(x)/k}.$$

However, without direct access to $\pi_{\text{ref}}$, we use the empirical frequency $\widehat{\pi}$ instead of $\pi_{\text{ref}}$ in the construction of $\widehat{\mathcal{Y}}_\alpha$, making $\widehat{\mathcal{Y}}_\alpha$ an *approximation* of $\mathcal{Y}_\alpha$. To extend the two properties of $\mathcal{Y}_\alpha$ to $\widehat{\mathcal{Y}}_\alpha$, we require the following event that sandwiches $\widehat{\mathcal{Y}}_{3/(4C^*(x))}(x)$ with $\mathcal{Y}_{1/C^*(x)}(x)$ and $\mathcal{Y}_{1/(4C^*(x))}(x)$:

$$\mathcal{E} : \mathcal{Y}_{1/C^*(x)}(x) \subset \widehat{\mathcal{Y}}_{3/(4C^*(x))}(x) \subset \mathcal{Y}_{1/(4C^*(x))}(x).$$

Under event $\mathcal{E}$, $\alpha$ can be set as $1/(4C^*(x))$ in (5.1). The complete expectation formula gives

$$\text{Regret}(x) = \mathbb{E}\left[r^*(x, y^*) - \max_{i \in [k]} r^*(x, \widetilde{y}_i)\Big|\mathcal{E}\right] \cdot \mathbb{P}(\mathcal{E}) + \mathbb{E}\left[r^*(x, y^*) - \max_{i \in [k]} r^*(x, \widetilde{y}_i)\Big|\neg\mathcal{E}\right] \cdot \mathbb{P}(\neg\mathcal{E})$$

$$\leq \epsilon_{\text{opt}}(x) + \sqrt{4C^*(x)\epsilon_{\text{RM}}^2(x)/k} + \mathbb{P}(\neg\mathcal{E}),$$

so it remains to characterize the probability of $\mathcal{E}$.

The probability of $\mathcal{Y}_{1/C^*(x)}(x) \not\subset \widehat{\mathcal{Y}}_{3/(4C^*(x))}(x)$ can be characterized by first bounding $\mathbb{P}(y \notin \widehat{\mathcal{Y}}_{3/(4C^*(x))}(x))$ for any $y \in \mathcal{Y}_{1/C^*(x)}(x)$ using the Chernoff bound, and then applying the union bound with the crucial observation of $|\mathcal{Y}_{1/C^*(x)}(x)| \leq C^*(x)$. When characterizing $\mathbb{P}(\widehat{\mathcal{Y}}_{3/(4C^*(x))}(x) \not\subset \mathcal{Y}_{1/(4C^*(x))}(x))$, we can similarly use the Chernoff bound in $\mathbb{P}(y \in \widehat{\mathcal{Y}}_{3/(4C^*(x))}(x))$ for any $y \in \mathcal{Y}(x)\backslash\mathcal{Y}_{1/(4C^*(x))}(x)$, but the union bound does not hold because the cardinality of the set $\mathcal{Y}(x)\backslash\mathcal{Y}_{1/(4C^*(x))}(x)$ is unknown. To resolve this issue, we assign elements of $\mathcal{Y}(x)\backslash\mathcal{Y}_{1/(4C^*(x))}(x)$ into "bins" $\{G_j\}$, each with capacity $1/(2C^*(x))$, i.e., $\pi_{\text{ref}}(G_j|x) \leq 1/(2C^*(x))$. The smallest number of bins is no more than $4C^*(x)$ because any two bins with $\pi_{\text{ref}}(G_j|x) \leq 1/(4C^*(x))$ can be merged. With this assignment, we can bound $\mathbb{P}(G_j \cap \widehat{\mathcal{Y}}_{3/(4C^*(x))}(x) \neq \varnothing)$ with the Chernoff bound, and then use the union bound with the bins, which resolves the problem because the number of bins is bounded. $\square$

## 6 GENERAL LOWER BOUNDS

In this section, we establish a lower bound that highlights the fundamental factors influencing the Pass@$k$ inference problem. Specifically, the bound depends on the coverage coefficient $C^*(x)$, the reward model estimation error $\epsilon_{\text{RM}}^2(x)$ and $\epsilon_{\text{opt}}(x)$, and the number of candidates $k$. It matches the upper bound in Theorem 5.1, which indicates that the algorithm BoM is minimax optimal.

**Theorem 6.1.** For a given prompt $x$, assume that $C^*(x) \geq 2k$. Then for any algorithm $\mathcal{A}$ for the Pass@$k$ inference problem, there exists an instance $\mathcal{I} = (\mathcal{X}, \mathcal{Y}, \pi^*, r^*, \pi_{\mathrm{ref}}, \widehat{r})$ such that the coverage coefficient is $C^*(x)$, and $(r^*, \widehat{r})$ satisfies Assumptions 3.1 and 3.2. Moreover, and regret can be lower bounded by

$$\mathrm{Regret}(x) = \Omega\Big(\epsilon_{\mathrm{opt}}(x) + \sqrt{C^*(x)\epsilon_{\mathrm{RM}}^2(x)/k}\Big).$$

Theorem 6.1 shows that the term $\epsilon_{\mathrm{opt}}(x)$ is unavoidable in the Pass@$k$ inference problem and does not diminish as the number of candidates $k$ increases. In contrast, the component associated with the expected squared loss, $\epsilon_{\mathrm{RM}}(x)$, decreases at a rate of $1/\sqrt{k}$. This bound matches the upper bound for BoM (Theorem 5.1), demonstrating that BoM is minimax optimal.

## 7 EXPERIMENTS

In this section, we empirically verify the effectiveness of our proposed BoM algorithm on mathematical reasoning tasks.

### 7.1 EXPERIMENT SETUP

**Models and Datasets.** We use Qwen3-4B-Instruct-2507 (Team, 2025) (Qwen3-4B) as the reference policy $\pi_{\mathrm{ref}}$[1]. We adopt AceMath-7B-RM (Liu et al., 2024) as the reward model $\widehat{r}$, a mathematical reward model trained on a large corpus generated by different language models which is selected due to its strong performance and moderate size. We adopt the widely used GSM8K (Cobbe et al., 2021), MATH-500 (Hendrycks et al., 2021), and AIME24[2] dataset as our testing corpus. We first sample $N$ trajectories and call the reward model to evaluate each trajectory. The answers are then extracted from the trajectories and clustered by mathematical equivalence[3]. For each answer group, we use the average of the rewards of all the corresponding trajectories as the reward of this group. We also calculate the frequency of each answer group as an estimation of $\pi_{\mathrm{ref}}(\cdot)$.

**Method and Baselines.** Given a specific $k$, we consider our method BoM, and two baselines, majority voting and BoN. In BoM, we set a threshold $\alpha$ and select the $k$ answers (up to mathematical equivalence) with highest reward score and frequency greater than $\alpha$. In BoN, we directly select the $k$ answers (up to mathematical equivalence) with highest rewards. As for majority voting, we directly select $k$ answers (up to mathematical equivalence) with highest frequency. Additionally, we consider the SBoN algorithm studied in Aminian et al. (2025); Verdun et al. (2025). Given $N$ generated answers $\{y_i\}_{i=1}^N$, we select $k$ answers in a sequence, without replacement according to a softmax distribution $p(y_i|x) \propto \exp(\beta\widehat{r}(x, y_i))$, where $\beta$ is a tunable parameter. We examine two variants: with and without reward calibration, where the calibration is implemented in the same way as in (Balashankar et al., 2024). In the calibrated case, rewards are normalized using the win rate over the reference policy $\pi_{\mathrm{ref}}$. We denote these versions SBoN(C) and SBoN, respectively.

### 7.2 RESULTS

**Results with varying $k$.** We first plot the results for $k \in \{1, 2, 3, 5, 10\}$ in Figure 1(a) for GSM8K, Figure 1(b) for MATH-500, and Figure 1(c) for AIME24. We sample $N = 2000$ for GSM8k, and $N = 500$ for MATH-500 and AIME24. On MATH-500, the performance of BoM consistently outperforms the baselines. On GSK8K and AIME24, BoM also shows a large improvement over majority voting and outperforms BoN for small $k$. These results empirically verify the effectiveness of the BoM algorithm.

**Results with varying $N$.** We also study the performance of the three methods under different sample sizes. We conduct the experiments on the AIME24 dataset and vary $N$ between 100 and 2000, with $k = 1, 3, 5$. Except for the case of $N = 100$ where the threshold of BoM is set to $\alpha = 0.015$, we use $\alpha = 0.005$ in all other settings. We compile the results in Figure 2. the performance of majority voting remains consistently low, which aligns with Theorem 4.1, demonstrating that majority voting incurs constant regret and does not benefit from increased sample size. The performance of BoN tends to degrade as $N$ increases. In contrast, when $N \geq 200$, BoM consistently outperforms both

---

[1]Please see Appendix F for results on additional models.

[2]https://huggingface.co/datasets/di-zhang-fdu/AIME_1983_2024

[3]The equivalence is determined through the standard implementation in Qwen2.5-Math repository and refers the readers to their public implementation for more details. https://github.com/QwenLM/Qwen2.5-Math/blob/a45202bd16f1ec06f433442dc1152d0074773465/evaluation/grader.py#L73

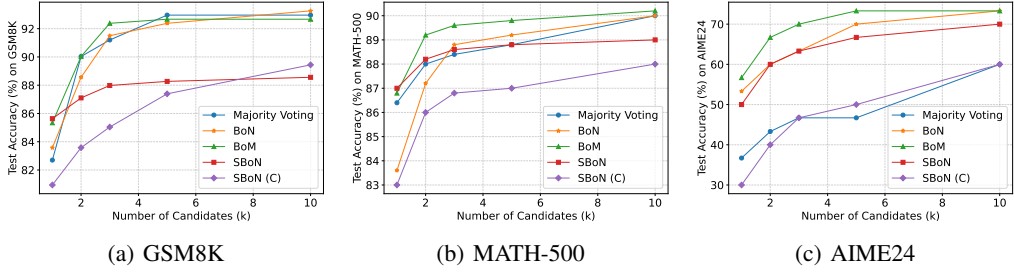

(a) GSM8K        (b) MATH-500        (c) AIME24

Figure 1: The results with different $k$. BoM consistently outperforms the baselines on MATH-500 for all $k$ and on AIME24, GSM8K when $k$ is small, and matches the performance of baselines in other settings.

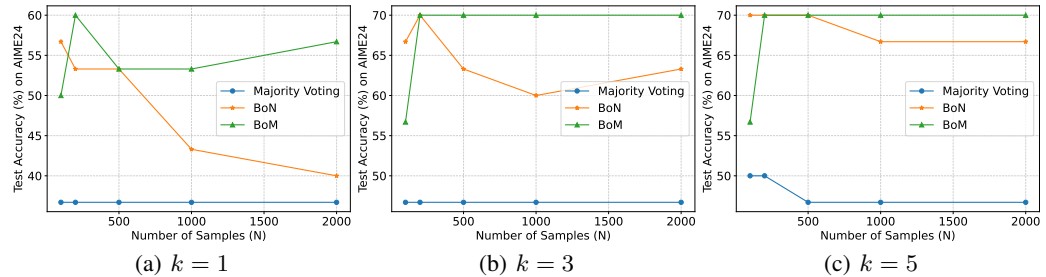

(a) $k = 1$        (b) $k = 3$        (c) $k = 5$

Figure 2: The results with fixed $k$ and different $N$. When $N$ increases, the performance of BoN is likely to decrease over all the $k$. The performance of Majority voting remains at a low level. Among them, BoM has a more consistent performance and outperforms baselines with larger $N$.

baselines and does not decrease significantly with the increase of $N$. This observation is consistent with our theoretical results, as BoM is scaling-monotonic.

### 7.3 Ablation study of $\alpha$

In this section, we present an ablation study of the hyperparameter $\alpha$ in our BoM algorithm. When $\alpha = 0$, BoM will degrade to BoN, meaning that as $\alpha$ approaches 0, BoM behaves similarly to BoN. We further observe that selecting a larger $\alpha$ can potentially improve the performance when $k$ is small, although this comes at the cost of deteriorated performance for larger $k$. More experimental results are included in Appendix D.

Table 2: Ablation study of $\alpha$ of BoM in MATH500, Qwen3-4B

| Pass@$k$ | 1 | 2 | 3 | 5 | 10 |
|---|---|---|---|---|---|
| $\alpha = 0$ (BoN) | 83.6 | 87.2 | 88.8 | 89.2 | 90 |
| $\alpha = 0.003$ | 86 | 88.8 | 89 | 89.2 | 90 |
| $\alpha = 0.005$ | 86.8 | 89.2 | 89.6 | 89.8 | 90.2 |
| $\alpha = 0.007$ | 86.6 | 89 | 89.6 | 89.8 | 90.2 |
| $\alpha = 0.011$ | 86.6 | 88.8 | 89.4 | 89.4 | 89.6 |
| $\alpha = 0.015$ | 87.4 | 88.8 | 89.4 | 89.4 | 89.8 |
| Majority voting | 86.4 | 88 | 88.4 | 88.8 | 90 |

## 8 Conclusion

In this work, we demonstrate the scaling laws of the Pass@$k$ inference problem by displaying the minimax lower bound of the regret and proposing the algorithm BoM with regret matching the lower bound. We also show that BoM has the advantage of scaling monotonicity compared with majority voting and BoN, which makes BoM preferable when scaling up the generation budget. For future work, it would be interesting to explore instance-dependent lower bounds, instead of the minimax lower bound as Theorems 4.1, 4.2 and 6.1. In addition, our current analysis assumes the uniqueness of the optimal response, such that the coverage coefficient will not depend on the choice of optimal policy $\pi^*$. How to define generalized coverage coefficients under the multiple optimum setting remains to be explored.

ACKNOWLEDGMENT

We thank the anonymous reviewers and area chair for their helpful comments. QD, KJ, XL, HZ and QG are supported in part by the National Science Foundation DMS-2323113, CPS-2312094, IIS-2403400 and the Sloan Research Fellowship. QD is also supported in part by Amazon PhD Fellowship. The views and conclusions contained in this paper are those of the authors and should not be interpreted as representing any funding agencies.

ETHICS STATEMENT

Our work investigates a novel Pass@$k$ inference problem, focusing on the theoretical analysis of different inference strategies. In addition, we propose a new algorithm, Best-of-Majority (BoM), which achieves optimal theoretical guarantees, and we further provide empirical validation to support its effectiveness. Importantly, our experiments focus on solving mathematical problems with LLMs, and the language models do not generate or promote harmful content, nor does it raise issues related to discrimination, bias, or fairness.

REPRODUCIBILITY STATEMENT

In this paper, we conduct experiments with open-source LLMs on widely used mathematical datasets. A detailed description of the models and datasets is provided in Section 7.1, while the key experimental parameters are discussed in Section 7.2. On the theoretical side, we present a proof sketch of the upper bound of the BoM algorithm in Section 5, with the complete proof deferred to Appendix B. Appendix C contains several lower-bound results, corresponding to the theorems in Sections 4.1, 4.2, and 6.

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

## A    COMPARISON WITH AMINIAN ET AL. (2025)

In Aminian et al. (2025), the theoretical guarantees of BoN are studied under a different setting, and a variant called SBoN (Soft BoN) is also analyzed. In this section, we provide a comprehensive comparison of the assumptions and theoretical regimes considered in both works. Aminian et al. (2025) made three main assumptions.

- The reward function is bounded, i.e., $0 \leq r^*(\cdot) \leq R_{\max}$, $0 \leq \widehat{r}(\cdot) \leq R_{\max}$.
- The reward estimation error $\epsilon_{\beta,r}(x) := \frac{1}{\beta} \log(\mathbb{E}_{\pi_{\mathrm{ref}}}[\exp(\beta(r^*(x,y) - \widehat{r}(x,y))^2)])$.
- Maximal reward can be achieved for the estimated reward $\widehat{r}$.

In comparison, the first assumption aligns with our setting with $R_{\max} = 1$. The second assumption aligns with our Assumption 3.1 only when $\beta = 0$. For the third assumption, we assume the maximal value can be achieved only for the true reward model $r^*$, with a unique optimizer. And the error at the optimal point is specially considered.

**Results of BoN**: Aminian et al. (2025) proved a regret of

$$\mathrm{Regret}(x) = \sqrt{\epsilon_{\infty,r}(x)}\left(\sqrt{C_{\infty,\widehat{r},\mathrm{ref}}(x)} + \sqrt{C_{\infty,r^*,\mathrm{ref}}(x)}\right) + c\sqrt{\log\left(1 + \frac{C_{\infty,\widehat{r},\mathrm{ref}}(x) - 1}{N}\right)},$$

for some constant $c > 0$. In this result, it considers two coverage definitions dependent on $\widehat{r}$ and $r^*$. Moreover, the reward estimation error is $\beta = \infty$, which corresponds to the case of supreme norm, instead of the squared norm. When $N \to \infty$, the convergence rate is $\sqrt{1/N}$.

*Comparison with our results*: One central distinction lies in our different focuses. Our work focuses on analyzing algorithmic performance within the Pass@$k$ framework, and therefore our theoretical results explicitly account for the ability to submit $k$ different answers, which is not studied in Aminian et al. (2025). In particular, for the theory of BoN, Theorem 4.2 establishes a **lower bound** that depends explicitly on $k$, which serves as a parallel result to Aminian et al. (2025).

As for upper bound of BoM, we prove that it has an exponential decay dependence of $N$, together with a term of $O(\sqrt{C\epsilon_{\mathrm{RM}}^2/k})$, which directly has $k$ dependence. Although the definition of reward error differs, thus the results can not be directly compared, we have shown a faster convergence rate

of $N$ than Aminian et al. (2025).

**Results of SBoN**: For SBoN, Aminian et al. (2025) proved a regret of

$$\text{Regret}(x) = \sqrt{\epsilon_{\beta,r}(x)}\Big(\sqrt{C_{\infty,\widehat{r},\text{ref}}(x)} + \sqrt{C_{\infty,r^*,\text{ref}}(x)}\Big)$$
$$+ c\sqrt{\log\Big(1 + \frac{C_{\infty,\widehat{r},\text{ref}}(x) - 1}{N}\Big)} + \log(C_{\infty,r^*,\text{ref}}(x))/\beta.$$

Note that $\beta \neq 0$, as the last term will explode. Thus, the setting of Aminian et al. (2025) is never same as ours. Again, when $N \to \infty$, the convergence rate of SBoN is $\sqrt{1/N}$, compared with our exponential decay. Finally, our focus is on the Pass@$k$ setting, which has never been analyzed in Aminian et al. (2025).

# B  THEORETICAL GUARANTEE OF BoM (ALGORITHM 3)

In this section, we will prove Theorem 5.1, which provides the theoretical upper bound of Algorithm 3. To start with, for any $\alpha > 0$, we denote

$$\mathcal{Y}_\alpha(x) = \{y \in \mathcal{A}(x) : \pi_\text{ref}(y|x) \geq \alpha\},$$

indicating the set of responses with relatively high probability for $\pi_\text{ref}$. Using the definition of the coverage coefficient (3.2), we have $y^* \in \mathcal{Y}_\alpha(x)$ as long as $\alpha \geq 1/C^*(x)$. Next, we will build the relationship between the empirical set $\widehat{\mathcal{Y}}_\alpha(x)$ and $\mathcal{Y}_\alpha(x)$. Denote $\mathcal{E}$ as the event such that

$$\mathcal{Y}_{1/C^*(x)}(x) \subset \widehat{\mathcal{Y}}_{3/(4C^*(x))} \subset \mathcal{Y}_{1/(4C^*(x))}(x).$$

Our proof consists of two parts:

**Step 1:** We first show that $\mathcal{E}$ holds with high probability.

**Step 2:** Provided that $\mathcal{E}$ holds, since $y^* \in \mathcal{Y}_{1/C^*(x)}(x)$, we have $y^* \in \widehat{\mathcal{Y}}_{3/(4C^*(x))}$; furthermore, since $\widetilde{y}_i \in \widehat{\mathcal{Y}}_{3/(4C^*(x))}$, we have $\widetilde{y}_i \in \mathcal{Y}_{1/(4C^*(x))}(x)$, so $\pi_\text{ref}(\widetilde{y}_i|x) \geq 1/(4C^*(x))$ for every submitted response $\widetilde{y}_i$. We can then characterize $\Delta_i = |r^*(x, \widetilde{y}_i) - \widehat{r}(x, \widetilde{y}_i)|$ using the definition of the estimation error $\epsilon_\text{RM}^2$. If $y^* \in \{\widetilde{y}_1, \ldots, \widetilde{y}_k\}$, then the regret is zero; if $y^* \notin \{\widetilde{y}_1, \ldots, \widetilde{y}_k\}$, then using Assumption 3.2, we have

$$r^*(x, y^*) - r^*(x, \widetilde{y}_i) \leq \underbrace{|r^*(x, y^*) - \widehat{r}(x, y^*)|}_{\epsilon_\text{opt}(x)} + \underbrace{[\widehat{r}(x, y^*)) - \widehat{r}(x, \widetilde{y}_i)]}_{\leq 0} + \underbrace{|\widehat{r}(x, \widetilde{y}_i) - r^*(x, \widetilde{y}_i)|}_{\Delta_i}.$$

Combining these parts together, we complete the proof of Theorem 5.1.

We now get into the details of the proof. The following lemma states that the event of $\mathcal{E}$ will occur with high probability:

**Lemma B.1.** $\mathcal{E}$ holds with probability at least $1 - 5C^*(x)e^{-N/(32C^*(x))}$.

*Proof.* The proof consists of two parts that characterize the probabilities of $\mathcal{Y}_{1/C(x)}(x) \not\subset \widehat{\mathcal{Y}}_{3/(4C^*(x))}$ and $\widehat{\mathcal{Y}}_{3/(4C^*(x))} \not\subset \mathcal{A}_{1/(4C^*(x))}(x)$, respectively:

**Part I: Probability of $\mathcal{Y}_{1/C^*(x)}(x) \not\subset \widehat{\mathcal{Y}}_{3/(4C^*(x))}$.** We first fix any $y \in \mathcal{Y}_{1/C^*(x)}(x)$. By Chernoff bound, we have

$$\mathbb{P}\big(\widehat{\pi}(y) < 3/(4C^*(x))\big) \leq \exp\Big(-\frac{N\pi_\text{ref}(y|x)}{2}\Big(1 - \frac{3}{4C^*(x)\pi_\text{ref}(a|x)}\Big)^2\Big) \leq e^{-N/(32C^*(x))}, \tag{B.1}$$

where the first inequality holds due to the Chernoff bound, and the second inequality holds because $\pi_\text{ref}(y|x) \geq 1/C^*(x)$. Applying the union bound to all $y \in \mathcal{Y}_{1/C^*(x)}(x)$, we have

$$\mathbb{P}\big(\mathcal{Y}_{1/C^*(x)}(x) \not\subset \widehat{\mathcal{Y}}_{3/(4C^*(x))}\big) = \mathbb{P}\Big(\bigvee_{y \in \mathcal{Y}_{1/C^*(x)}(x)} \mathbb{1}[\widehat{\pi}(y) \leq 3/(4C^*(x))]\Big)$$
$$\leq \sum_{y \in \mathcal{Y}_{1/C^*(x)}(x)} \mathbb{P}\big(\widehat{\pi}(y) < 3/(4C^*(x))\big)$$
$$\leq 1 - |\mathcal{Y}_{1/C^*(x)}(x)| \cdot e^{-N/(32C^*(x))}$$

$$\leq 1 - C^*(x)e^{-N/(32C^*(x))}, \tag{B.2}$$

where the first inequality holds due to the union bound, the second inequality holds due to (B.1), and the last inequality holds because $|\mathcal{Y}_{1/C^*(x)}(x)| \leq C^*(x)$.

**Part II: Probability of $\widehat{\mathcal{Y}}_{3/(4C^*(x))} \not\subset \mathcal{A}_{1/(4C^*(x))}(x)$.** We cannot use the same union bound as (B.2) because the cardinal of the set to take union bound $\mathcal{Y}\backslash\mathcal{Y}_{1/(4C^*(x))}(x)$ is unknown. To resolve this issue, we first partition $\mathcal{Y}\backslash\mathcal{Y}_{1/(4C^*(x))}(x)$ into groups, then apply Chernoff bound to each group, and finally apply the union bound to the groups. This technique resolves the problem because the number of groups is in the order of $\mathcal{O}(C^*(x))$, and the union bound goes through without incurring the cardinality of $\mathcal{Y}\backslash\mathcal{Y}_{1/(4C^*(x))}(x)$.

In detail, suppose that $\mathcal{Y}\backslash\mathcal{Y}_{1/(4C^*(x))}(x) = \{y_i\}_{i\geq 1}$. We start with a single group $G_1 = \varnothing$, and add $y_i$ to one of the groups sequentially. For each response $y_i \in \mathcal{Y}\backslash\mathcal{Y}_{1/(4C^*(x))}(x)$, if there exists group $G_j$ such that

$$\pi_{\text{ref}}(y_i|x) + \sum_{y\in G_j} \pi_{\text{ref}}(y|x) \leq \frac{1}{2C^*(x)}, \tag{B.3}$$

then we update $G_j$ with $G_j \cup \{a_i\}$ where $j$ is the smallest index that satisfies (B.3). Otherwise, we create a new group $\{a_i\}$. From the construction of the groups, we can easily see that the probability of any group $G_j$ under the reference model satisfies

$$\pi_{\text{ref}}(G_i|x) = \sum_{a\in G_j} \pi_{\text{ref}}(a|x) \leq \frac{1}{2C^*(x)}. \tag{B.4}$$

Furthermore, the total number of groups $M$ should be no larger than $4C^*(x)$ because otherwise, suppose that (B.3) does not holds for $y_i$ and any existing group $G_j(j \in [M])$ where $M > 4C^*(x) - 1$, i.e.,

$$\sum_{y\in G_j} \pi_{\text{ref}}(y|x) > \frac{1}{2C^*(x)} - \pi_{\text{ref}}(y_i|x) > \frac{1}{4C^*(x)}, \tag{B.5}$$

where the last inequality holds because $\pi_{\text{ref}}(a) < 1/(4C^*(x))$. We then have

$$1 = \sum_{y\in\mathcal{Y}} \pi_{\text{ref}}(y|x)$$

$$\geq \left[\pi_{\text{ref}}(y_i|x) + \sum_{y\in G_1} \pi_{\text{ref}}(y|x)\right] + \sum_{j=2}^{M} \left[\sum_{y\in G_j} \pi_{\text{ref}}(y|x)\right]$$

$$\geq \frac{1}{2C^*(x)} + (M-1)\cdot\frac{1}{4C^*(x)}$$

$$> \frac{1}{2C^*(x)} + (4C^*(x) - 1 - 1)\cdot\frac{1}{4C^*(x)} = 1,$$

where the first inequality holds because the union of $a_i$ and all existing groups is a subset of $\mathcal{A}(x)$, the second inequality holds due to (B.5), and the last inequality holds due to the assumption of $M > 4C^*(x) - 1$. We have thus arrived at a contradiction, and we conclude that $M \leq 4C^*(x)$. For each group, we apply the Chernoff bound:

$$\mathbb{P}\left(\bigvee_{y\in G_j} \mathbb{1}[\widehat{\pi}(y) \geq 3/(4C^*(x))]\right)$$

$$\leq \mathbb{P}\big(\widehat{\pi}(G_j) \geq 3/(4C^*(x))\big)$$

$$\leq \exp\left(-N\frac{(3/(4C^*(x)) - \pi_{\text{ref}}(G_i|x))^2}{3/(4C^*(x)) + \pi_{\text{ref}}(G_i|x)}\right)$$

$$\leq e^{-N/(20C^*(x))}, \tag{B.6}$$

where the first inequality holds because if the frequency of one response in $G_j$ is larger than $3/(4C^*(x))$, then the total frequency of group $G_j$ should be larger than $3/(4C^*(x))$; the second

inequality holds due to the Chernoff bound; the last inequality holds due to (B.4). Applying the union bound to all groups,

$$\mathbb{P}\big(\widehat{\mathcal{Y}}_{3/(4C^*(x))} \not\subset \mathcal{A}_{1/(4C^*(x))}(x)\big) = \mathbb{P}\Big(\bigvee_{y \in \mathcal{Y}\setminus\mathcal{Y}_{1/(4C^*(x))}} \mathbb{1}[\widehat{\pi}(y) \geq 3/(4C^*(x))]\Big)$$

$$\leq \sum_{j=1}^{M} \mathbb{P}\Big(\bigvee_{y \in G_j} \mathbb{1}[\widehat{\pi}(y) \geq 3/(4C^*(x))]\Big)$$

$$\leq M e^{-N/(20C^*(x))}$$

$$\leq 4C^*(x) e^{-N/(32C^*(x))}, \tag{B.7}$$

where the first inequality holds due to the union bound, the second inequality holds due to (B.6), and the last inequality holds because $M \leq 4C^*(x)$ and $e^{-N/(20C^*(x))} \leq e^{-N/(32C^*(x))}$. Combining (B.2) and (B.7), using the union bound, we have

$$\mathbb{P}(\mathcal{E}) \geq 1 - 5C e^{-N/(32C^*(x))}.$$

Thus, we have completed the proof of Lemma B.1. $\qquad\square$

Using this lemma, we then proceed with the proof of Theorem 5.1:

*Proof of Theorem 5.1.* Suppose that $\mathcal{E}$ holds. If $y^*$ is included in the submitted responses, then the regret is $0$. We now consider the case where $y^*$ is not submitted. According to the definition of the coverage coefficient, we have

$$\pi_{\text{ref}}(y^*|x) \geq \pi^*(y^*|x)/C^*(x) \geq 1/C^*(x),$$

so $y^* \in \mathcal{Y}_{1/C^*(x)}(x)$. Furthermore, since $\mathcal{Y}_{1/C^*(x)}(x) \subset \widehat{\mathcal{Y}}_{3/(4C^*(x))}$ when $\mathcal{E}$ holds, we have $y^* \in \widehat{\mathcal{Y}}_{3/(4C^*(x))}$. Since $y^*$ is not selected as the output, we know that (i) at least $k$ responses are submitted because otherwise all responses in $\widehat{\mathcal{Y}}_{3/(4C^*(x))}$ would be submitted, and (ii) $\widehat{r}(x, y^**) \leq \widehat{r}(x, \widetilde{y}_i)$ for any $i \in [k]$. We thus have

$$\widehat{r}(x, \widetilde{y}_i) \geq \widehat{r}(x, y^*) \geq r^*(x, y^*) - \epsilon_{\text{opt}}(x), \tag{B.8}$$

where the second inequality holds due to Assumption 3.2. Therefore, the regret conditioned on event $\mathcal{E}$ is

$$\min_{i \in [k]}\{r^*(x, y^*) - r^*(x, \widetilde{y}_i)\} \leq \epsilon_{\text{opt}}(x) + \min_{i \in [k]}\{\widehat{r}(x, \widetilde{y}_i) - r_*(x, \widetilde{y}_i)\}$$

$$\leq \epsilon_{\text{opt}}(x) + \sqrt{\frac{1}{k}\sum_{i=1}^{k}|\widehat{r}(x, \widetilde{y}_i) - r_*(x, \widetilde{y}_i)|^2}$$

$$\leq \epsilon_{\text{opt}}(x) + \sqrt{\frac{4C^*(x)}{k}\sum_{i=1}^{k}\pi_{\text{ref}}(\widetilde{y}_i|x)|\widehat{r}(x, \widetilde{y}_i) - r^*(x, \widetilde{y}_i)|^2}$$

$$\leq \epsilon_{\text{opt}}(x) + \sqrt{\frac{4C^*(x)}{k}\sum_{y \in \mathcal{Y}}\pi_{\text{ref}}(y|x)|\widehat{r}(x, y) - r^*(x, y)|^2}$$

$$= \epsilon_{\text{opt}}(x) + \sqrt{\frac{4C^*(x)\epsilon_{\text{RM}}^2(x)}{k}}, \tag{B.9}$$

where the first inequality holds due to (B.8), the second inequality holds because the minimum is no larger than the average, the third inequality holds because $\pi_{\text{ref}}(y|x) \geq 1/(4C^*(x))$ for any $y \in \widehat{\mathcal{Y}}_{3/(4C^*(x))}$ when $\widehat{\mathcal{Y}}_{3/(4C^*(x))} \subset \mathcal{Y}_{1/(4C^*(x))}(x)$, the fourth inequality holds because $\{\widetilde{y}_1, \ldots, \widetilde{y}_k\}$ is a subset of $\mathcal{Y}$, and the last equality holds due to the definition of the estimation error $\epsilon_{\text{RM}}^2(x)$. Combining (B.9) with the case where $y^* \in \{\widetilde{y}_1, \ldots, \widetilde{y}_k\}$ and the regret is $0$, we conclude that under condition $\mathcal{E}$,

$$r^*(x, y^*) - \max_{i \in [k]} r^*(x, \widetilde{y}_i) \leq \epsilon_{\text{opt}}(x) + \sqrt{\frac{4C^*(x)\epsilon_{\text{RM}}^2(x)}{k}}. \tag{B.10}$$

Finally, we take the complete expectation of the regret:

$$\text{Regret}(x) = \mathbb{E}\left[r^*(x, y^*) - \max_{i \in [k]} r^*(x, \widetilde{y}_i) \Big| \mathcal{E}\right] \cdot \mathbb{P}(\mathcal{E}) + \mathbb{E}\left[r^*(x, y^*) - \max_{i \in [k]} r^*(x, \widetilde{y}_i) \Big| \neg\mathcal{E}\right] \cdot \mathbb{P}(\neg\mathcal{E})$$

$$\leq \left(\epsilon_{\text{opt}}(x) + \sqrt{\frac{4C^*(x)\epsilon_{\text{RM}}^2(x)}{k}}\right) \cdot \mathbb{P}(\mathcal{E}) + 1 \cdot \mathbb{P}(\neg\mathcal{E})$$

$$\leq \epsilon_{\text{opt}}(x) + \sqrt{\frac{4C^*(x)\epsilon_{\text{RM}}^2(x)}{k}} + 5C^*(x)e^{-N/(32C^*(x))},$$

where the first inequality holds due to (B.10) and $\text{Regret}(x) \leq 1$, and the second inequality holds because $\mathbb{P}(\mathcal{E}) \leq 1$ and due to Lemma B.1. Finally, when $N \geq 16C^*(x) \log\left(kC^*(x)/\epsilon_{\text{RM}}^2(x)\right)$, we have

$$\text{Regret}(x) \leq \epsilon_{\text{opt}}(x) + O\left(\sqrt{C^*(x)\epsilon_{\text{RM}}^2(x)/k}\right).$$

We complete the proof of Theorem 5.1.

$\square$

## C  PROOF OF LOWER BOUNDS

In this section, we will prove the lower bounds used in the main text of this paper. Specifically, we establish the results for majority voting (Theorem 4.1), Best-of-$N$ (Theorem 4.2), and the general case of Pass@$k$ inference algorithms (Theorem 6.1). Before proceeding, we first establish an independent lower bound regarding $\epsilon_{\text{opt}}(x)$. This result is general and can be applied to any subsequent lower bound, introducing an additional $\epsilon_{\text{opt}}(x)$ term.

### C.1  LOWER BOUND REGARDING $\epsilon_{\text{OPT}}(x)$

We first study the following hard case where any algorithm for the Pass@$k$ inference problem suffers from the regret of $\Omega(\epsilon_{\text{opt}}(x))$. Combining this lower bound with any algorithm-dependent lower bound $b$ (obtained from the analysis of a hard instance), we can show that the lower bound of the algorithm is

$$\Omega(\max\{\epsilon_{\text{opt}}(x), b\}) = \Omega(\epsilon_{\text{opt}}(x) + b).$$

**Lemma C.1.** Assume that $\epsilon_{\text{opt}}(x) \leq \sqrt{C^*(x)\epsilon_{\text{RM}}^2(x)}$ and $C^*(x) \geq 2k$. Then there exists an instance $\mathcal{I} = (\mathcal{X}, \mathcal{Y}, \pi^*, r^*, \pi_{\text{ref}}, \widehat{r})$ such that the coverage coefficient is $C^*(x)$, and $(r^*, \widehat{r})$ satisfy Assumptions 3.1 and 3.2. Furthermore, for any prompt $x \in \mathcal{X}$, the regret of any algorithm for the Pass@$k$ inference problem satisfies

$$\text{Regret}(x) = \Omega(\epsilon_{\text{opt}}(x)).$$

*Proof.* For simplicity, we omit the prompt $x$ in our proof. We apply the idea of the averaging hammer, which considers a total of $M$ hard instances such that no algorithm can perform well on all instances. It is a technique commonly used in the proof of lower bounds (see e.g., Theorem 24.1 in Lattimore & Szepesvári (2020)). The responses set is $\{y_0, y_1, \ldots, y_M\}$ for all $M$ hard instances. The reference policy and the approximate reward model are also shared by all instances:

$$\pi_{\text{ref}}(y_0) = 1 - M/C^*, \quad \pi_{\text{ref}}(y_1) = \cdots \pi_{\text{ref}}(y_M) = 1/C^*;$$
$$\widehat{r}(y_0) = 0, \quad \widehat{r}(y_1) = \cdots = \widehat{r}(y_M) = 1 - \epsilon_{\text{opt}}.$$

The hard instances are different only in the ground-truth reward model and $\pi^*$. For instance $\mathcal{I}_j = (\mathcal{X}, \mathcal{Y}, \pi_j^*, r_j^*, \widehat{r}, \pi_{\text{ref}})$ where $j \in [M]$, we set

$$\pi_j^*(y_i) = \delta_{ij}, \quad r_j^*(y_i) = \begin{cases} 0 & i = 0; \\ 1 & i = j; \\ 1 - \epsilon_{\text{opt}} & \text{otherwise.} \end{cases}$$

For all hard cases, the total estimation error is $\epsilon_{\text{opt}}^2/C* \leq \epsilon_{\text{RM}}^2$. Among these $M$ hard instances, any algorithm that outputs up to $k$ responses will fail to output the optimal response in at least $M - k$ instances, inducing the regret of $\epsilon_{\text{opt}}$. Therefore, the average regret of these $M$ instances is at least

$$\text{Regret} \geq \frac{M - k}{M}\epsilon_{\text{opt}}.$$

Setting $M = 2k$, we have $\text{Regret} = \Omega(\epsilon_{\text{opt}})$. $\square$

## C.2 LOWER BOUND OF MAJORITY VOTING (THEOREM 4.1)

*Proof of Theorem 4.1.* For simplicity, we omit the prompt $x$ in our proof. Consider the following hard instance. The size of the response set is $2 + k$, with $\mathcal{Y} = \{y_0, y^*, y_1, y_2, \dots, y_k\}$. The ground truth reward satisfies:

$$r^*(y_0) = 0; \qquad r^*(y^*) = 1; \qquad r^*(y_i) = 1/2, \quad \forall 1 \le i \le k.$$

Therefore, the optimal policy $\pi^*$ satisfies:

$$\pi^*(y_0) = 0; \qquad \pi^*(y^*) = 1; \qquad \pi^*(y_i) = 0, \quad .$$

In this instance, we assume that the estimated reward function $\widehat{r}$ is accurate. Let $\eta = 2w(1)/w(1/2)$. We further define the reference policy as:

$$\pi_{\text{ref}}(y_0) = 1 - (1 + \eta k)/C^*; \qquad \pi_{\text{ref}}(y^*) = 1/C^*; \qquad \pi_{\text{ref}}(y_i) = \eta/C^*, \quad \forall 1 \le i \le k.$$

The reference polity is well defined as long as $C^* \ge 1 + 2kw(1)/w(1/2)$. Now we consider the sampled responses $\widehat{y}_1, \widehat{y}_2, \dots, \widehat{y}_N$. Define

$$N^* = \sum_{j=1}^{N} \mathbb{1}(\widehat{y}_j = y^*); \qquad N_i = \sum_{j=1}^{N} \mathbb{1}(\widehat{y}_j = y_i), \quad \forall i \in [k].$$

Then the expectations of $N^*$ and $N_i$ are

$$\mathbb{E}[N^*] = \frac{N}{C^*}; \qquad \mathbb{E}[N_i] = \frac{\eta N}{C^*}, \quad \forall 1 \le i \le k.$$

Using the Chernoff bounds, we have

$$\mathbb{P}\Big[\frac{N^*}{N} \ge \frac{3}{2C^*}\Big] \le \exp\Big(\frac{-N}{9C^*}\Big), \quad \mathbb{P}\Big[\frac{N_i}{N} \le \frac{3\eta}{4C^*}\Big] \le \exp\Big(\frac{-N\eta}{4C^*}\Big). \tag{C.1}$$

Denote $\mathcal{E}$ as the event such that

$$\frac{N^*}{N} \le \frac{3}{2C^*}; \qquad \frac{N_i}{N} \ge \frac{3\eta}{4C^*}, \quad \forall i \in [k].$$

Taking the union bound with (C.1), we have

$$\mathbb{P}(\mathcal{E}) \ge 1 - \exp\Big(\frac{-N}{9C^*}\Big) - k\exp\Big(\frac{-N\eta}{4C^*}\Big) \ge 1 - (k+1)\exp\Big(\frac{-N}{9C^*}\Big),$$

where the last inequality holds because $\eta > 1$. Under event $\mathcal{E}$, we have

$$\frac{w(1/2)N_i}{w(1)N^*} = \frac{N_i/N}{N^*/N} \cdot \frac{w(1/2)}{w(1)} \ge \frac{3\eta/(4C^*)}{3/(2C^*)} \cdot \frac{2}{\eta} = 1,$$

where the inequality holds due to the definition of the event $\mathcal{E}$ and the definition of $\eta$. Therefore, conditioned on event $\mathcal{E}$, the (weighted) majority voting (Algorithm 1) will output $\{y_1, \dots, y_k\}$ and suffer from a 1/2 regret. To summarize, the regret satisfies

$$\text{Regret} \ge \mathbb{P}(\mathcal{E}) \cdot \mathbb{E}[\text{Regret}|\mathcal{E}] \ge \frac{1}{2}\Big(1 - (k+1)\exp\Big(\frac{-N}{9C^*}\Big)\Big).$$

When $N \ge 9C^*(x)\log(2k+2)$,

$$1 - (k+1)\exp\Big[\frac{-N}{9C^*}\Big] \ge 1/2.$$

$\square$

## C.3 LOWER BOUND OF BON (THEOREM 4.2)

To prove Theorem 4.2, we construct two hard instances to accommodate two cases: (i) When $N$ is small, then it is very likely that $y^*$ does not even appear in $\{\widehat{y}_1, \dots, \widehat{y}_N\}$; (ii) When $N$ is large, then it is very likely that a number of responses that are suboptimal in $r^*$ but better than $y^*$ in $\widehat{r}$ are sampled. The two hard instances share the same structure but are different in parameters.

*Proof of Theorem 4.2.* For simplicity, we omit the prompt $x$. We consider two hard instances, one for $N \leq C^*$ and the other for $N \geq C^*$.

**Case 1:** $N \leq C^*$. We consider a hard instance with $\mathcal{Y} = \{y_0, y^*\}$, and

$$\pi^*(y_0) = 0, \quad \pi^*(y^*) = 1; \qquad r^*(y_0) = 0, \quad r^*(y^*) = 1;$$
$$\pi_{\text{ref}}(y_0) = 1 - 1/C^*, \quad \pi_{\text{ref}}(y^*) = 1/C^*; \qquad \widehat{r}(y_0) = 0, \quad \widehat{r}(y^*) = 1.$$

For this instance, the estimation errors are $\epsilon_{\text{opt}} = \epsilon_{\text{RM}} = 0$. If no sample in $\widehat{y}_1, \ldots, \widehat{y}_N$ is $y^*$, then the regret is 1. The probability that $y^* \notin \{\widehat{y}_1, \ldots, \widehat{y}_N\}$ is $(1 - 1/C^*)^N$. Therefore, we have

$$\text{Regret} \geq (1 - 1/C^*)^N \geq (1 - 1/C^*)^{C^*} \geq 1/4,$$

where the second inequality holds because $N \leq C^*$, and the second inequality holds because $C^* \geq 2$. Therefore, the BoN algorithm incurs constant regret in this hard instance when $N \leq C^*$.

**Case 2:** $N \geq C^*$. We consider the following hard instance: The response set is $\mathcal{Y} = \{y^*, y_0, y_1, \ldots, y_M\}$. Let $p > 0$ be a parameter to be determined. The reward models are

$$r^*(y^*) = 1, \quad r^*(y_0) = 0, \quad r^*(y_i) = 1 - \frac{\epsilon_{\text{RM}}}{2\sqrt{p}};$$
$$\widehat{r}(y^*) = 1 - \delta, \quad \widehat{r}(y_0) = 0, \quad \widehat{r}(y_i) = 1.$$

where $\delta < \epsilon_{\text{opt}}$ is a sufficiently small positive number to ensure that the reward of $y_1, \ldots, y_M$ is slightly better than $y^*$ in $\widehat{r}$, but $y^*$ is still the optimal response in $r^*$. In this way, $\pi^*(y^*) = 1$ and $\pi^*(y_i) = 0$ for $i = 0, 1, \ldots, M$. The reference model satisfies

$$\pi_{\text{ref}}(y^*) = 1/C^*, \quad \pi_{\text{ref}}(y_0) = 1 - 1/C^* - p, \quad \pi_{\text{ref}}(y_i) = p/M.$$

For this instance, the coverage is $C^*$, and the estimation error is less than $\epsilon_{\text{RM}}^2$ when $\delta$ is sufficiently small.

**Simple analysis.** We first consider a simple setting where $M = k$. When $\widehat{y}_1, \ldots, \widehat{y}_N$ covers every response in $\{y_1, \ldots, y_k\}$, then $\{y_1, \ldots, y_k\}$ will be the output of BoN, causing the regret of $\epsilon_{\text{RM}}/2\sqrt{p}$. The probability of any $y_i$ not being covered is

$$(1 - p/k)^N.$$

Using the union bound, the probability that there exists $y_i$ not being coverer is upper bounded by

$$\mathbb{P}[\exists i, y_i \notin \{\widehat{y}_1, \ldots, \widehat{y}_N\}] \leq k(1 - p/k)^N.$$

Thus, the regret of making the wrong decisions in $y_1, \ldots, y_k$ is lower bounded by

$$1 - k(1 - p/k)^N.$$

Then the regret satisfies

$$\text{Regret} \geq \left(1 - k(1 - p/k)^N\right) \cdot \frac{\epsilon_{\text{RM}}}{2\sqrt{p}}. \tag{C.2}$$

In this instance, when $\sqrt{N\epsilon_{\text{RM}}^2/[k\log(2k)]}/2 < 1$, we select $p = (k/N) \cdot \log(2k)$. Then, $\epsilon_{\text{RM}}/(2\sqrt{p}) = \sqrt{N\epsilon_{\text{RM}}^2/[k\log(2k)]}/2 < 1$. Thus, the constructed $r^*$ satisfies $0 \leq r^*(\cdot) \leq 1$. Then we have

$$1 - k(1 - p/k)^N = 1 - k\left(1 - \frac{\log(2k)}{N}\right)^N$$
$$\geq 1 - k\left[\exp\left(-\frac{\log(2k)}{N}\right)\right]^N$$
$$= 1 - k\exp\left(-\log(2k)\right)$$
$$= 1/2,$$

where the first inequality holds due to the basic inequality $1 - x \leq \exp(-x), \forall x \in \mathbb{R}$. Substituting this into (C.2), we have proved that the regret can be lower bounded by $\Omega(\sqrt{N\epsilon_{\text{RM}}^2/(k\log k)})$.

Otherwise, when $\sqrt{N\epsilon_{\text{RM}}^2/[k\log(2k)]}/2 \geq 1$, let $p = \epsilon_{\text{RM}}^2/4$. Then, the regret in (C.2) can be lower bounded by

$$\text{Regret} \geq \left(1 - k\left(1 - \frac{\epsilon_{\text{RM}}^2}{4k}\right)^N\right)$$

$$\geq \left(1 - k\left(1 - \frac{\epsilon_{\text{RM}}^2}{4k}\right)^{4k\log(2k)/\epsilon_{\text{RM}}^2}\right)$$

$$\geq \left(1 - k\left[\exp\left(-\frac{\epsilon_{\text{RM}}^2}{4k}\right)\right]^{4k\log(2k)/\epsilon_{\text{RM}}^2}\right)$$

$$= 1 - k\exp\left(-\log(2k)\right)$$

$$= 1/2,$$

where the third inequality holds due to the basic inequality $1 - x \leq \exp(-x), \forall x \in \mathbb{R}$. Therefore, we have

$$\text{Regret} \geq \Omega\left(\min\left\{1, \sqrt{N\epsilon_{\text{RM}}^2/(k\log k)}\right\}\right).$$

This analysis will lead to an additional logarithmic term on $k$, which is unnecessary. To avoid this term, we consider the following improved analysis.

**Improved analysis.** We consider the instance where $M = 2k$. Consider the event where at least $k$ responses among $y_1, \ldots, y_M$ are covered by $\widehat{y}_1, \ldots, \widehat{y}_N$. Since $\widehat{r}(y_i) > \widehat{r}(y^*)$ for $i = 1, \ldots, M$, the optimal responses $y^*$ is not included in $\widetilde{y}_1, \ldots, \widetilde{y}_k$, which also incurs the regret of $\epsilon_{\text{RM}}/(2\sqrt{p})$. We now consider the probability of this event. Define the following random variables:

- Define $S$ as the number of samples within $y_1, \ldots, y_M$, i.e.,

$$S = \sum_{i=1}^{N}\sum_{j=1}^{M}\mathbb{1}[\widehat{y}_i = y_j].$$

- Define $O_j$ as the occupancy of $y_j$, i.e.,

$$O_j = \bigvee_{i=1}^{N}\mathbb{1}[\widehat{y}_i = y_j].$$

- Define $D$ as the total occupancy of $\{y_1, \ldots, y_M\}$, i.e.,

$$D = \sum_{j=1}^{M}O_j.$$

Our goal is to lower bound $\mathbb{P}(D \geq k)$. Fix $s_0 > k$. Using the total expectation formula, we have

$$\mathbb{P}(D \geq k) = \sum_{s \geq k}\mathbb{P}(D \geq k|S = s)\mathbb{P}(S = s)$$

$$\geq \sum_{s \geq s_0}\mathbb{P}(D \geq k|S = s)\mathbb{P}(S = s)$$

$$\geq \mathbb{P}(D \geq k|S = s_0)\mathbb{P}(S \geq s_0), \tag{C.3}$$

where the first inequality holds because $s_0 \geq k$, and the second inequality holds because $\mathbb{P}(D \geq k|S = s) \geq \mathbb{P}(D \geq k|S = s_0)$ when $s \geq s_0$. We then calculate the two probabilities separately. We first use the Chernoff bound to characterize $\mathbb{P}(S \geq s_0)$. The expectation of $S$ is

$$\mathbb{E}[S] = \sum_{i=1}^{N}\mathbb{P}(\widehat{y}_i \in \{y_1, \ldots, y_M\}) = Np.$$

Then by the Chernoff bound, we have

$$\mathbb{P}(S \geq s_0) \geq 1 - \exp\left(-\frac{(Np - s_0)^2}{2Np}\right). \tag{C.4}$$

We then calculate the conditional probability $\mathbb{P}(D \geq k | S = s_0)$, and we assume without loss of generality that $\widehat{y}_1, \ldots, \widehat{y}_{s_0}$ fall within $\{y_1, \ldots, y_M\}$. Conditioned on this event $\mathcal{E}$, we have $\mathbb{P}(\widehat{y}_i = y_j) = 1/M$ for $1 \leq i \leq s_0$ and $1 \leq j \leq M$. Although we cannot use the vanilla Chernoff bound to bound $\mathbb{P}(D \geq k | S = s)$, we can use the Chernoff bound for **negatively-correlated** random variables (Dubhashi & Ranjan, 1996) (or See Theorem 4.3 in Dubhashi & Panconesi (1998)) to bound the probability. We first calculate the expectation of $D$, which is

$$\mathbb{E}[D | S = s_0] = M\mathbb{E}[O_j] = M(1 - \mathbb{P}[\widehat{y}_i \neq y_j, \forall i \in [s_0]]) = M(1 - (1 - 1/M)^{s_0}).$$

We then verify that $O_1, \ldots, O_M$ are negatively correlated, which is to show that for any subset $\mathcal{J} \subset [M]$, we have $\mathbb{E}[\prod_{j \in \mathcal{J}} O_j] \leq \prod_{j \in \mathcal{J}} \mathbb{E}[O_j]$, i.e., $\mathbb{P}(O_j = 1, \forall j \in \mathcal{J}) \leq \prod_{j \in \mathcal{J}} \mathbb{P}(O_j = 1)$. We prove by induction with respect to the cardinality of $\mathcal{J}$. The inequality is trivial When $|\mathcal{J}| = 1$. Suppose that the inequality holds for all $\mathcal{J}$ such that $|\mathcal{J}| \leq n$. It then suffices to show the inequality holds for $\mathcal{J} = [n + 1]$. Note that

$$\mathbb{P}(O_1 = 1, \ldots, O_{n+1} = 1)$$
$$= \mathbb{P}(O_1 = 1, \ldots, O_n = 1) - \mathbb{P}(O_1 = 1, \ldots, O_n = 1 | O_{n+1} = 0) \cdot \mathbb{P}(O_{n+1} = 0)$$
$$= \mathbb{P}(O_1 = 1, \ldots, O_n = 1) \cdot \mathbb{P}(O_{n+1} = 1)$$
$$+ \big[\mathbb{P}(O_n = 1, \ldots, O_n = 1) - \mathbb{P}(O_1 = 1, \ldots, O_n = 1 | O_{n+1} = 0)\big] \cdot \mathbb{P}(O_{n+1} = 0),$$

Using the induction hypothesis, we have

$$\mathbb{P}(O_1 = 1, \ldots, O_n = 1) \cdot \mathbb{P}(O_{n+1} = 1) \leq \prod_{j=1}^{n+1} \mathbb{P}(O_j = 1).$$

It then suffices to show that

$$\mathbb{P}(O_n = 1, \ldots, O_n = 1) \leq \mathbb{P}(O_1 = 1, \ldots, O_n = 1 | O_{n+1} = 0),$$

which is trivial because the event $\widehat{y}_i = y_j (j \in [n])$ becomes more likely conditioned of the event that $\widehat{y}_i \neq y_{n+1}$. Therefore, the inequality holds for $|\mathcal{J}| = n + 1$, and we complete the verification of $O_j$ being negatively correlated. Therefore, using the Chernoff bound for negatively-correlated random variables, we have

$$\mathbb{P}(D \geq k | S = s_0) \geq 1 - \exp\left(-\frac{\{M[1 - (1 - 1/M)^{s_0}] - k\}^2}{2M[1 - (1 - 1/M)^{s_0}]}\right). \tag{C.5}$$

Substituting (C.4) and (C.5) into (C.3), we have

$$\text{Regret} \geq \mathbb{P}(D \geq k) \cdot \frac{\epsilon_{\text{RM}}}{2\sqrt{p}}$$

$$\geq \frac{\epsilon_{\text{RM}}}{2\sqrt{p}} \cdot \left[1 - \exp\left(-\frac{\{M[1 - (1 - 1/M)^{s_0}] - k\}^2}{2M[1 - (1 - 1/M)^{s_0}]}\right)\right] \cdot \left[1 - \exp\left(-\frac{(Np - s_0)^2}{2Np}\right)\right]. \tag{C.6}$$

Let $M = 2k, s_0 = 3k$. If $\sqrt{N\epsilon_{\text{RM}}^2/k}/4 \leq 1$, we set $p = 4k/N$. Then, $\epsilon_{\text{RM}}/(2\sqrt{p}) = \sqrt{N\epsilon_{\text{RM}}^2/k}/4 < 1$. Thus, the constructed $r^*$ satisfies $0 \leq r^*(\cdot) \leq 1$. In this case, we have

$$1 - (1 - 1/M)^{s_0} = 1 - \left(1 - \frac{1}{2k}\right)^{3k} \geq 1 - e^{-1.5} \geq \frac{3}{4}.$$

We thus have

$$1 - \exp\left(-\frac{\{M[1 - (1 - 1/M)^{s_0}] - k\}^2}{2M[1 - (1 - 1/M)^{s_0}]}\right)$$
$$\geq 1 - \exp\left(-\frac{(2k \cdot 3/4 - k)^2}{2 \cdot 2k \cdot 3/4}\right)$$
$$= 1 - e^{-k/12} \geq 1 - e^{-1/12},$$

where the second inequality holds because $k \geq 1$. We also have $Np = 4k$, so

$$1 - \exp\left(-\frac{(Np - s_0)^2}{2Np}\right) = 1 - \exp\left(-\frac{(4k - 3k)^2}{2 \cdot 4k}\right) = 1 - e^{-k/8} \geq 1 - e^8,$$

where the last inequality holds because $k \geq 1$. Combining all the above, we have

$$\text{Regret} \geq \frac{\epsilon_{\text{RM}}}{\sqrt{4k/N}} \cdot (1 - e^{-1/12}) \cdot (1 - e^{-1/8}) \geq 0.004 \sqrt{\frac{N\epsilon_{\text{RM}}^2}{k}}.$$

Otherwise, if $\sqrt{N\epsilon_{\text{RM}}^2/k}/4 \geq 1$ let $p = \epsilon_{\text{RM}}^2/4$. Then, with the same argument as that in the Simple analysis part, the regret is lower bounded by $\Omega(1)$. Therefore, we have

$$\text{Regret} \geq \Omega\Big(\min\Big\{1, \sqrt{N\epsilon_{\text{RM}}^2/k}\Big\}\Big).$$

$\square$

## C.4  GENERAL LOWER BOUND (THEOREM 6.1)

We first provide a more general version of Theorem 6.1:

**Theorem C.2.** Assume that $C^*(x) \geq \max\{k, 2\}$. Then for any positive integer $M \in [k, C^*(x)]$ and any algorithm $A$ that outputs $k$ responses, there exists a hard instance $\mathcal{I} = (\mathcal{X}, \mathcal{Y}, \pi^*, r^*, \pi_{\text{ref}}, \widehat{r})$ such that the coverage is $C$, the estimation error is $\epsilon_{\text{RM}}^2$, and the regret of algorithm $A$ satisfies

$$\text{Regret}(x) \geq \frac{M - k}{M} \sqrt{\frac{C^*(x)\epsilon_{\text{RM}}^2}{M - 1}}.$$

When $C \geq 2k$, we can set $M = 2k$ and obtain the regret lower bound of $\Omega(\sqrt{C\epsilon_{\text{RM}}^2/k})$ in Theorem 6.1. We now present the proof of Theorem C.2.

*Proof of Theorem C.2.* We consider the case of $\mathcal{X} = \{x\}$, and omit the prompt $x$ in $A(x)$, $\pi_{\text{ref}}(\cdot|x)$, $\widehat{r}(x, \cdot)$, etc.

To prove Theorem 6.1, we again apply the idea of the averaging hammer, and consider a total of $M$ hard instances such that no algorithm can perform well on all instances. All of these hard instances have a total of $M + 1$ possible responses $\mathcal{Y} = \{y_0, \ldots, y_M\}$, and we aim to make $y_1, \ldots, y_M$ hard to distinguish from each other. In detail, all hard instances also share the same reference model and the same $\widehat{r}$:

$$\pi_{\text{ref}}(y_0) = 1 - M/C, \quad \pi_{\text{ref}}(y_1) = \cdots = \pi_{\text{ref}}(y_M) = 1/C;$$
$$\widehat{r}(y_0) = 0, \quad \widehat{r}(y_1) = \cdots = \widehat{r}(y_M) = 1.$$

For hard instance $\mathcal{I}_j (j \in [M])$, we make $y_j$ the optimal response with ground truth reward being 1 and $\pi^*(y_j) = 1$, and make all other responses suboptimal with a gap of $\delta$, i.e., $\mathcal{I}_j = (\mathcal{X}, \mathcal{Y}, \pi_j^*, r_j^*, \pi_{\text{ref}}, \widehat{r})$, where

$$\pi_j^*(y_l) = \delta_{jl}, \quad r_j^*(y_l) = \begin{cases} 0 & l = 0; \\ 1 & l = j; \\ 1 - \delta & \text{otherwise.} \end{cases}$$

In this hard instance, the coverage is $C$, and in order to make the estimation error equal to $\epsilon_{\text{RM}}^2$, we require

$$(M - 1) \cdot \delta^2 \cdot 1/C = \epsilon_{\text{RM}}^2,$$

which indicates that $\delta = \sqrt{C\epsilon_{\text{RM}}^2/(M - 1)}$. Since any algorithm can only output a maximum of $k$ different responses, it cannot output the optimal response in at least $M - k$ out of the $M$ hard instances, suffering from the regret of at least $\delta$. Therefore, the averaged regret of the $M$ instances is at least

$$\frac{1}{M}\sum_{j=1}^{M} \mathbb{E}_{\widetilde{y}_1, \ldots, \widetilde{y}_k \sim A}\big[r_j^*(y_j) - \max\big\{r_j^*(\widetilde{y}_1), \cdots, r_j^*(\widetilde{y}_k)\big\}\big] \geq \frac{1}{M} \cdot (M - k) \cdot \delta = \frac{M - k}{M}\sqrt{\frac{C\epsilon_{\text{RM}}^2}{M - 1}}.$$

Therefore, there exists an instance $\mathcal{I}_{j^*}$ within the $M$ hard instances such that

$$\mathbb{E}_{\widetilde{y}_1, \ldots, \widetilde{y}_k \sim A}\big[r_{j^*}^*(y_{j^*}) - \max\big\{r_{j^*}^*(\widetilde{y}_1), \cdots, r_{j^*}^*(\widetilde{y}_k)\big\}\big] \geq \frac{M - k}{M}\sqrt{\frac{C\epsilon_{\text{RM}}^2}{M - 1}}.$$

$\square$

## D FURTHER ABLATION STUDIES

In this section, we present additional ablation studies on the choice of $\alpha$, using the AIME24 and GSM8K datasets.

Table 3: Ablation study of $\alpha$ of BoM in AIME24, Qwen3-4B

| Pass@$k$ | 1 | 2 | 3 | 5 | 10 |
|---|---|---|---|---|---|
| $\alpha = 0$ (BoN) | 53.3 | 60 | 63.3 | 70 | 73.3 |
| $\alpha = 0.003$ | 56.7 | 66.7 | 70 | 73.3 | 73.3 |
| $\alpha = 0.005$ | 53.3 | 66.7 | 70 | 70 | 70 |
| $\alpha = 0.007$ | 56.7 | 70 | 70 | 70 | 70 |
| $\alpha = 0.011$ | 50 | 63.3 | 63.3 | 63.3 | 63.3 |
| $\alpha = 0.015$ | 50 | 60 | 60 | 60 | 60 |
| Majority voting | 36.7 | 43.3 | 46.7 | 46.7 | 60 |

Table 4: Ablation study of $\alpha$ of BoM in GSM8k, Qwen3-4B

| Pass@$k$ | 1 | 2 | 3 | 5 | 10 |
|---|---|---|---|---|---|
| $\alpha = 0$ (BoN) | 83.58 | 88.56 | 91.50 | 92.38 | 93.26 |
| $\alpha = 0.003$ | 85.34 | 90.32 | 92.38 | 92.67 | 92.67 |
| $\alpha = 0.005$ | 85.34 | 90.03 | 92.08 | 92.38 | 92.38 |
| $\alpha = 0.007$ | 85.92 | 90.62 | 92.08 | 92.08 | 92.08 |
| $\alpha = 0.011$ | 86.22 | 90.32 | 91.50 | 91.50 | 91.50 |
| $\alpha = 0.015$ | 86.22 | 90.32 | 91.20 | 91.20 | 91.20 |
| Majority voting | 82.70 | 90.03 | 91.20 | 92.96 | 92.96 |

## E ABLATION STUDY OF REWARD MODEL

To further demonstrate that our findings are not tied to any particular reward model, we additionally evaluate using InternLM2-reward (Cai et al., 2024). This model is not specifically trained for mathematical reasoning and shows weaker performance in the evaluation of Liu et al. (2024). Even under this weaker reward model, our algorithm behaves consistently and leads to the same overall conclusion.

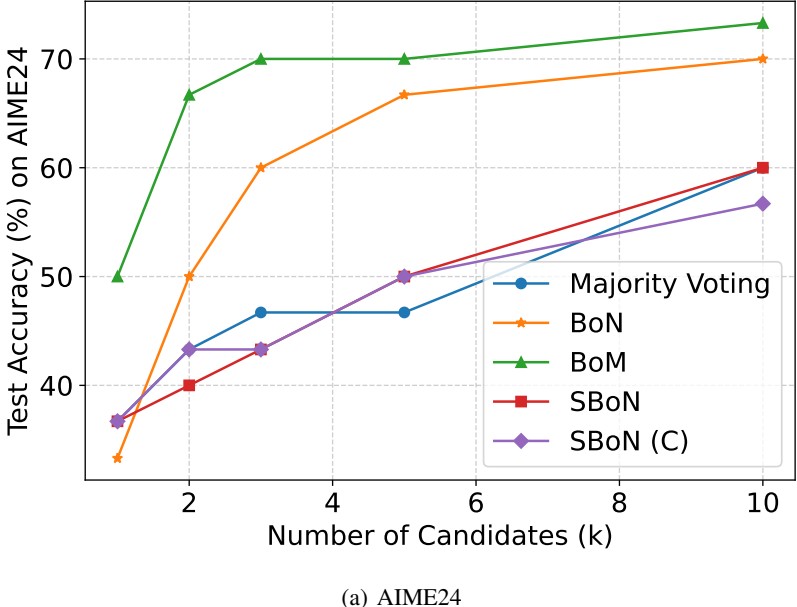

(a) AIME24

Figure 3: The results of different $k$ with $N = 500$ using InternLM2-reward, Qwen3-4B-Instruct .

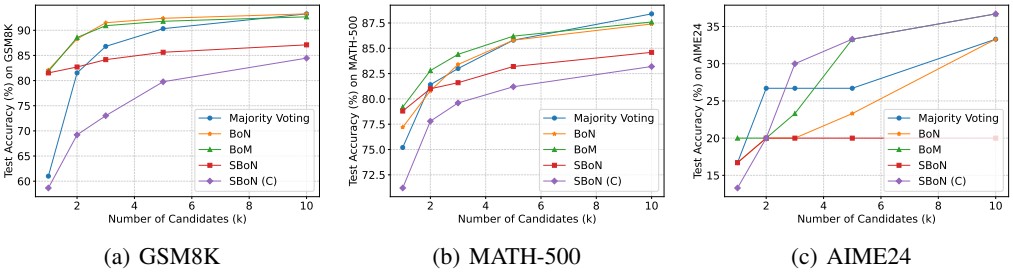

| (a) GSM8K | (b) MATH-500 | (c) AIME24 |

Figure 4: The results of different $k$ with $N = 500$ on Qwen2.5-1.5B.

# F ADDITIONAL EXPERIMENTS

In this section, we conduct experiments on an additional model, Qwen2.5-Math-1.5B-Instruct (Qwen2.5-1.5B) for more results. The other experiment setups follows the experiments on Qwen3 unless specified. The results on Qwen2.5-1.5B are compiled in Figure 4. In particular, BoM matches the performance of BoN on GSM8k and outperforms BoN on MATH-500 and AIME24. The performance of BoM also surpasses majority voting on GSM8k and MATH-500 with $k \leq 5$. These results shows that BoM demonstrates a better overall performance over baselines when $k$ is small.

## THE USE OF LARGE LANGUAGE MODELS (LLMs)

We use LLMs as a tool to refine our writing and correct grammatical errors.

