# OpenReview forum: "Best-of-Majority: Minimax-Optimal Strategy for Pass@k Inference Scaling"
_ICLR.cc/2026/Conference — ICLR 2026 Poster_

### Official Review · Reviewer_Df6F · 2025-10-25

**Soundness:** 3
**Presentation:** 3
**Contribution:** 3
**Rating:** 6
**Confidence:** 4

**Summary:**

The paper considers scaling laws for inference: A model is required to output $k$ responses $y$ to a given prompt $x$, and the regret, compared to the optimal response, is measured according to the best one in this set of $k$ responses. In this paper the model is allowed to output a total of $N$ responses, and pass chosen $k$ ones to evaluation, based on a given estimated reward model. Thus, $N$ represents the computational cost of inference. A minimax lower bound on the regret as a function of $k$ (as well as the reward model estimation error) is stated, and the regret of two previously proposed algorithms (used in practice) Majority voting and Best-of-$N$) is analyzed. Both these algorithms are shown to be non-minimax. An algorithm (Best-of-Majority, BoM) is proposed which is shown to be minimax optimal with respect to $k$. The main idea of the algorithm is that even if a large number of responses $N$ are generated, they should be filtered to the most frequent ones, which are also assumed to be the ones in which the reward model is most reliable.

**Strengths:**

1) The paper derive theoretical bounds for a timely problem of inference scaling laws.

2) Two practical algorithms are analyzed, which are shown to be sub-optimal, and a new one is proposed, which is established to be minimax optimal.

**Weaknesses:**

1) The main goal of the paper is to study the trade-off between the computational cost of inference, as expressed by $N$ and the regret. From this aspect, it is unreasonable that both the upper for BoM and the minimax lower bound do not depend on $N$, and thus do not capture this trade-off.

2) Theorem 5.1: The result does not cover the regime $k\lesssim N\lesssim C^{*}(x)$.

3) The algorithms and analysis are based on empirical counts over the response alphabet $\mathcal{Y}$. It is not obvious how the performance scales with this alphabet size.

4) The proposed algorithm requires the knowledge of $C^*(x)$ in order to make sure that $N$ is large enough. Similarly, the algorithm requires knowledge of a lower bound on $\epsilon_{RM}$ which is somewhat cumbersome since the reward model is designed to have a small error.

**Questions:**

1) Line 82 - We further “introduce a formal definition of scaling-monotonicity”: What is exactly the contribution of this paper to this definition, beyond Huang et al. 2025). Line 188 - “we introduce the reference policy..”: The same question. 2) Could you explain the dependence on $C^{\star}$  and why it is essential? Under Assumption 3.2, $C^{\star}$ is the inverse of the probability of the reference policy at the optimal response. Isn't it possible to slightly mix $\pi_{ref}$ with a uniform distribution over the response alphabet, to make $C^{\star}$ effectively a constant?

3) Line 204 – “sufficiently many samples are observed” is unclear: Samples for reward estimation or responses?

4) Line 252- “each receiving higher probability under $\pi_{ref}$”: Higher than what?

5) “Moreover, our earlier analysis reveals complementary strengths of these methods...” - this is unclear since the analysis thus (Theorems 4.1 and 4.2) only focused on impossibility bounds.

6) In (B.4), I would propose to add a citation for the Chernoff bound for negatively correlated random variables.

7) In the proof of Theorem B.2, what is meant by “the idea of averaging hammer”

---

> ### Author Response · Authors · 2025-11-21
>
> Many thanks for your detailed review and positive feedback!
>
> ---
>
> **Q1**: The main goal of the paper is to study the trade-off between the computational cost of inference, as expressed by $N$ and the regret. It's unreasonable that both the upper for BoM and the minimax lower bound do not depend on $N$.
>
> **A1**: For Theorem 5.1, we would like to correct the misunderstanding of the reviewer: The regret upper bound of BoM contains a term dependent on $N$, but when choosing $N=\tilde\Omega(C^*(x))$, the term is dominated by others so that the Theorem 5.1 can be presented in a simpler form.
>
> In detail, we prove that the upper bound contains a term that decays exponentially in $N$, i.e., $\text{Regret}(x)\le \epsilon_{\text{opt}}(x) + O(\sqrt{C^\*(x) \epsilon_{\text{RM}}^2(x)/k}) + O(C^\*(x)\exp(-N/(32C^\*(x)))$ (Line 858) which stems from the analysis of the high-probability event. In our revision, we have illustrated the dependence of $N$ more explicitly in the main text (Lines 344-347).
>
> Theorem 6.1 is a general lower bound for **any** algorithm in the minimax sense. This lower bound holds even for algorithms that do not select responses from the sampled responses, for which the sampling budget $N$ does not exist. The closest related result in [1] is Proposition 2.3, which likewise does not include any dependence on $N$.
>
> ---
>
> **Q2**: Theorem 5.1 does not cover the regime $k \lesssim N \lesssim C^*$.
>
> **A2**: A similar argument as the first part of Theorem 4.2 can show that when $N \le C^\*(x)$, there exists a problem instance such that for any algorithm, the worst-case regret is $\Omega(1)$. In this case, We can define an instance with $\pi_{\text{ref}}(y^\*) = 1/C^\*$. Then with probability $(1-1/C^\* )^N \ge(1-1/C^\*)^{C^\*}=\Omega(1)$, the optimal answer will not be sampled, and no inference algorithm can select the optimal response $y^\*$. An important advantage of our Theorem 5.1 is that we only need $\tilde \Theta(C^\*)$ samples to guarantee our regret upper bound.
>
> ---
>
> **Q3**: The algorithms and analysis are based on empirical counts over the response alphabet $\mathcal{V}$. It is not obvious how the performance scales with this alphabet size.
>
> **A3**: In LLM generation tasks, the potential size of the response alphabet can be extremely large. For instance, in a math problem with a numerical answer, the solution can be any number in the range of the answer. A theoretical bound that depends on the alphabet size would therefore become vacuous or uninformative in such settings. For this reason, avoiding any dependence on the alphabet size is not a weakness. On the contrary, it is an important strength of our theorem, ensuring meaningful guarantees even when the action space is enormous.
>
> ---
>
> **Q4**:The proposed algorithm requires the knowledge of $C^*(x)$ in order to make sure that $N$ is large enough. Similarly, the algorithm requires knowledge of a lower bound on $\epsilon_{\text{RM}}$ which is somewhat cumbersome since the reward model is designed to have a small error.
>
> **A4**: First, a minor clarification: because our upper bound in Theorem 5.1 decays exponentially with $N$ (as discussed in A2), we can replace the $\epsilon(x)$ with any accuracy $\epsilon$ that we hope to achieve. Thus, the algorithm does not require knowledge of $\epsilon_{\text{RM}}$.
>
> For the dependence of $C^\*(x)$ of $N$, as we show in **A2**, we can prove that $\Omega(C^\*(x))$ samples are required for any algorithm. In the literature, even when $k=1$, [2] proved BoN will get optimal regret only when $N = \tilde \Theta(C^*(x))$. Thus, this problem-dependent term is unavoidable for inference-time algorithms, and not unique to our solution.
>
> ---
>
> **Q5**: What is exactly the contribution of this paper to this definition, beyond Huang et al. 2025)
>
> **A5**: In Huang et al. (2025), the property of scaling-monotonicity is described as "for all $N$ sufficiently large, and there is no risk of dropping below the theoretical bound as we scale computation". This reflects an intuitive property that with a good algorithm, increasing the computation should never be a bad thing. However, They do not rigorously define it in the mathematical language. As far as we know, we are the first to strictly define this property. Therefore, our contribution is to make the formal definition, instead of proposing a completely new concept.
>
> ---
>
> **Q6**: Line 204 – “sufficiently many samples are observed” is unclear: Samples for reward estimation or responses?
>
> **A6**: It means sufficient many samples of responses generated in the inference, corresponding to the condition ($\exists N_0 \text{ s.t. }N >N_0$). We have revised the argument to avoid any confusion.

---

> ### Author Response · Authors · 2025-11-21
>
> **Q7**: Line 252- “each receiving higher probability under $\pi_{\text{ref}}$”: Higher than what?
>
> **A7**: It should be each receiving higher probability under $\pi_{\text{ref}} than the optimal response.
>
> ---
>
> **Q8**: "Moreover, our earlier analysis reveals complementary strengths of these methods..." - this is unclear since the analysis thus (Theorems 4.1 and 4.2) only focused on impossibility bounds.
>
> **A8**: After each impossibility result, we explicitly describe how the corresponding hard instance is constructed (Lines 250–253 and 279–281). Thus, instances that do not satisfy the properties of these hard cases are precisely those on which the algorithms perform well. In our revision, we have clarified this explanation to make the intended message clearer.
>
> ---
>
> **Q9**: In (B.4), I would propose to add a citation for the Chernoff bound for negatively correlated random variables. In the proof of Theorem B.2, what is meant by "the idea of averaging hammer".
>
> **A9**: Thanks for your suggestion. The idea of averaging hammer means a standard technique used in the proof of bandit lower bound (See e.g. Theorem 24.1 in [1]).
>
> We have revised our paper according to your valuable suggestions.
>
> [1]  Bandit Algorithm, Lattimore, Tor and Szepesvari, Csaba, 2020
>
> [2] Is Best-of-N the Best of Them? Coverage, Scaling, and Optimality in Inference-Time Alignment, Huang et al. 2025

---

> > ### Comment · Reviewer_Df6F · 2025-11-26
> >
> > Thank you for the detailed response. Given these clarifications, I recommend acceptance.
> > One point which I still cannot follow is regarding Q4: The bounds depend on $C^*(x)$, but is it strictly necessary for the algorithm to know its value in advance? Can it somehow be learned on-the-fly?

---

> > > ### Author Response · Authors · 2025-11-26
> > >
> > > Thank you for your positive feedback. We will further clarify the role of $C ^{\*}$.
> > > The coverage coefficient is introduced primarily from a theoretical perspective, consistent with prior work such as [2]. In particular, it can be shown that any algorithm requires at least $\Omega(C^{\*})$ samples to avoid constant regret. In contrast, our theorem demonstrates that once $N \ge \tilde\Theta(C^{\*})$, our algorithm admits a good theoretical guarantee.
> > >
> > > In our theoretical analysis, two components depend on $C^{\*}$, the choice of $\alpha$, and the sample size $N$. Both can be chosen in practice in a reasonable manner without directly knowing the value of $C^*$.
> > >
> > > **Sample size $N$**: The restriction of $N \ge \tilde\Theta(C^*)$ serves only as a lower bound. In practice, $N$ can be chosen arbitrarily large. Thanks to the scaling-monotonic property, the guarantee remains valid.
> > >
> > > **Selection of $\alpha$**: $\alpha$ can be treated as a tunable hyperparameter. We can determine it using grid search and performance evaluation on a separate hold-out dataset. We additionally included an ablation study on the influence of $\alpha$ on the final performance (Section 7.3).

---

### Official Review · Reviewer_kb59 · 2025-10-30

**Soundness:** 3
**Presentation:** 2
**Contribution:** 2
**Rating:** 6
**Confidence:** 3

**Summary:**

This paper studies the Pass@k inference scaling problem for large language models (LLMs), where a model can generate $N$ candidate responses and select up to $k$ outputs, and only the best of them is used to compute the regret. The authors observe that widely used inference strategies for Pass@1 inference, such as majority voting and Best-of-$N$ (BoN), fail to exhibit desirable scaling behavior as $k$ and $N$ increase.

To address this, they propose a new inference strategy, Best-of-Majority (BoM), which ensures a $O(\epsilon_{\text{opt}} + \sqrt{C^{*}\epsilon_{\text{RM}}^2/k})$ regret bound. Here, $C^\star$ is the coverage constant, $\epsilon_{\text{RM}}$ is the estimation error of the reward model, and $\epsilon_{\text{opt}}$ is the estimation error of reward at the optimal response. They also provide a matching minimax lower bound for any Pass@k inference strategy. Importantly, BoM is shown to be scaling-monotonic, meaning its performance does not degrade with larger sampling budgets $N$, which separates the.

Empirically, the method is validated on GSM8K, MATH-500, and AIME24, where BoM consistently outperforms or matches the baselines.

**Strengths:**

1. Strong theoretical foundation: The authors show strong lower bounds on the regret of previous methods of majority vote and BoN, providing insightful explanations on why they do not benefit from the scaling law.
The construction of the BoM method, which combines the advantages of the two previous methods, is simple and intuitive. They also show matching lower and upper bounds to demonstrate the minimax optimality for BoM.
The theoretical model and analysis are novel.

2. Good empirical validation: Experiments on various datasets align well with theory and convincingly support the claims.

**Weaknesses:**

Practical relevance:

i. It is slightly unclear from the text the broader impact of the Pass@k inference problem. The authors do mention the usage of Pass@k alignment in training LLMs in Section 2, but it could be beneficial to discuss its application in LLM inference or generation.

ii. In the paper, the BoM method is proved to be scaling monotonic, which is emphasized as a major separation from previous methods. From my understanding, this property guarantees that the error should converge to zero eventually as $N$ increases. However, in the experiments, the BoM method doesn't seem to benefit from this property, e.g., in Figure 2(a), the accuracy decreases at first, and in Figures 2(b) and 2(c), the accuracy is always $0.7$. This seems to suggest that we shouldn't use large $N$ in practice anyway. Is it because of the inherent error from the chosen reward model, i.e., $\epsilon_{\text{RM}}$, or is $N$ not large enough?

Clarity:

i. The authors could provide more explanation of the intuition of the technical terms, e.g., the coverage constant, the definitions in Assumption 3.1 and Assumption 3.2, especially for audiences with diverse backgrounds.

ii. In Theorem 5.1, the parameter $\alpha$ of the BoM algorithm is chosen to be $\Omega(1/C^*)$. In the experiment, the authors set $\alpha = 0.015$ for $N = 100$ and $\alpha = 0.005$ for other choices of $N$. I would assume the coverage constant is hard to compute in practice, but the authors do not explain the reasons for their choices of $\alpha$. Why are different parameters needed for different $N$? Also, I'm curious about how the accuracy varies with different $\alpha$.

**Questions:**

Please refer to the 'Weaknesses' section.

---

> ### Author Response · Authors · 2025-11-21
>
> We appreciate your positive review and suggestions on improving the clarity!
>
> ---
>
> **Q1**: It is slightly unclear from the text the broader impact of the Pass@k inference problem. The authors do mention the usage of Pass@k alignment in training LLMs in Section 2, but it could be beneficial to discuss its application in LLM inference or generation.
>
> **A1**: We believe the reviewer has missed a key part of importance in our work. First, Pass@$k$ is itself a core metric for evaluating LLM generation quality. Therefore, studying this problem directly and designing an optimal algorithm for Pass@$k$ is inherently valuable, which has well discussed in our introduction part (Lines 52-73). Apart from being this, we mention Pass@$k$ also influences other components of the LLM pipeline, including training.
>
> ---
>
> **Q2**: In the paper, the BoM method is proved to be scaling monotonic, which is emphasized as a major separation from previous methods. From my understanding, this property guarantees that the error should converge to zero eventually as $N$ increases.
>
> **A2**: We would like to correct a misunderstanding of the reviewer: the definition of scaling monotonicity does not guarantees that the regret should converge to zero eventually as $N\to\infty$. Instead, it holds only when **the reward model is also accurate enough**: "there exists $\epsilon_0>0$ and $N_0\in\mathbb{N}\_+$ such that for any $N\ge N_0$ and any instance that satisfies Assumption 3.1 with $\epsilon_{\text{RM}}(x)\le\epsilon_0$". Here, the accuracy of the reward model is also required.
>
> ---
>
> **Q3**: However, in the experiments, the BoM method doesn't seem to benefit from this property, e.g., in Figure 2a, the accuracy decreases at first, and in Figures 2b and 2c, the accuracy is always 0.7. This seems to suggest that we shouldn't use large $N$ in practice anyway. Is it because of the inherent error from the chosen reward model, i.e., $\epsilon_{\text{RM}}$, or is $N$ not large enough?
>
> **A3**: As we have addressed in **A2**, our theory does not say the regret will converge to 0 when $N$ is large enough. Instead, according to Theorem 5.1, when $N$ is large enough, we have $\mathrm{Regret}(x) \le \epsilon_{\text{opt}}(x) + O\Big(\sqrt{{C^*(x)\epsilon_{\text{RM}}^2(x)}/{k}}\Big)$. Zero regret occurs only when the reward estimation error is 0, i.e., $\epsilon_{\mathrm{opt}}=\epsilon_{\mathrm{RM}}=0$. In fact, our lower-bound result (Theorem 6.1) demonstrates the existence of a fundamental performance threshold that cannot be surpassed—regardless of the algorithm used or how large $N$ is chosen.
>
> Empirically, we find that our method reaches this theoretical threshold faster than the baselines. Furthermore, the monotonicity property ensures that our algorithm continues to satisfy its theoretical guarantees as $N$ increases. We never claim that the performance of our algorithm will be strictly increasing as $N$ increases.
>
> The statement that *we shouldn't use large $N$ in practice anyway* is not true for scaling-monotonic algorithms like BoM. When $N$ is small, increasing $N$ will very likely improve the performance. When the performance has already achieved the threshold, increasing $N$ will not be as effective as before, even harmful for algorithms without scaling monotonicity, such as BoN. However, the performance of BoM does not degrade due to scaling monotonicity.
>
> ---
>
> **Q4**: The authors could provide more explanation of the intuition of the technical terms, e.g., the coverage constant, the definitions in Assumption 3.1 and Assumption 3.2, especially for audiences with diverse backgrounds.
>
> **A4**: Thanks for your suggestion. The coverage coeffient evaluates the performance of the reference model, while Assumptions 3.1 and 3.2 are about the estimation error of the reward models. We have provided more explanations in the revised manuscript to make the argument more accessible.
>
> ---
>
> **Q5**: In Theorem 5.1, the parameter of the BoM algorithm is chosen to be $1/C^*$. In the experiment, the authors set 0.015 for $N=100$ and 0.005 for other choices of $N$. I would assume the coverage constant is hard to compute in practice, but the authors do not explain the reasons for their choices of $\alpha$. Why are different parameters needed for different $N$? Also, I'm curious about how the accuracy varies with different $\alpha$.
>
> **A5**:  We always set $\alpha=0.005$ when $N\ge200$. For $N=100$, however, since $0.005 * 100 = 0.5 < 1$, BoM with $\alpha=0.005$ will reduce to BoN. Therefore we select another value of 0.015. We also provide the ablation study of $\alpha$ in our revision (Section 7.3). When $\alpha=0$, BoM will degrade to BoN, meaning that as $\alpha$ approaches 0, BoM behaves similarly to BoN. We further observe that selecting larger $\alpha$ can potentially improve the performance when $k$ is small, although this comes at the cost of deteriorated performance for larger $k$.

---

> ### Comment · Reviewer_kb59 · 2025-11-27
>
> Thank you for the detailed response. I want to further clarify some of my questions:
>
> Q1: I raised this question because the clarification on motivation (in the text) is not enough from my perspective. I noticed that in lines 52-73, the authors mentioned "While most existing analyses focus on inference algorithms that output a single response, there are tasks that allow for multiple candidate outputs, where it is considered solved if any one of them is correct." Could you be more specific about the tasks? I appreciate the authors' changes in Section 2.
>
> Q2/3: I understood that the scaling-monotonicity requires the assumption on the accuracy of the reward model (Indeed, this is why I asked "Is it because of the inherent error from the chosen reward model?" in Q3). I believe that it is important to prove theoretically the scaling-monotonicity, which shows a separation between BoM and BoN. I also admire your empirical results showing that BoM outperforms the baseline methods consistently.
>
> However, I raised Q3 because the empirical results **do not match** the intuition of scaling-monotonicity.
> Especially, in the response, the authors claimed "increasing $N$ will not be as effective as before, even harmful for algorithms without scaling monotonicity". However, in Figures 2(b) and 2(c), the performance of BoN does not degrade or even improves when $N$ moves from $1000$ to $2000$.
> If you want to illustrate "When $N$ is small, increasing $N$ will very likely improve the performance. ", maybe just focus on $N \le 500$.
> On the other hand, in Figure 2(a), the performance of BoM degrades when $N$ moves from $200$ to $500$, which is inconsistent with this claim.
> More clarification is needed in this section.
>
> I would be more convinced if these concerns were resolved.

---

> > ### Author Response · Authors · 2025-12-02
> >
> > Although there seems to be some discrepency between the theory and practice of scaling monotonicity, we would like to argue that the problem instances in practice can be different from the hard instances in theory. The experiments can also be subject to randomness in sampling. These two factors can contribute unexpected experiment results, but scaling monotonicity is still a useful guideline when choosing $N$.
> >
> > We hereby resolve the concerns of the reviewer in detail:
> >
> > > The authors claimed "increasing $N$ will not be as effective as before, even harmful for algorithms without scaling monotonicity". However, in Figures 2(b) and 2(c), the performance of BoN does not degrade or even improves when $N$ moves from 1000 to 2000.
> >
> > We proved that BoN is not scaling monotonic by constructing a hard instance (Theorem 4.2). This hard instance represents the case where the reward model $\hat r$ is inaccurate. There are also instances where the reward model is accurate, and the performance benefits from larger $N$. We believe that the practical dataset and the reward model constitute a mixture of hard instances and benign instances, which can explain the increased accuracy from $N=1000$ to $N=2000$.
> >
> > > If you want to illustrate "When $N$ is small, increasing $N$ will very likely improve the performance", maybe just focus on $N\le500$. On the other hand, in Figure 2(a), the performance of BoM degrades when $N$ moves from 200 to 500, which is inconsistent with this claim.
> >
> > We believe that the high accuracy of BoM at $N=200$ in Figure 2(a) is most likely an outlier, possibly due to the randomness of sampling. This phenomenon can also be attributed to the worst-case regret that is possibly loose under specific settings.
> >
> > Therefore, we believe that scaling monotonicity is a useful property that can provide guidance to the selection of $N$. For algorithms without the property of scaling monotonicity like BoN, careful finetuning of $N$ is required, but for our algorithm BoM, the performance does not degrade with larger $N$ in general, although there can be outliers due to data randomness.

---

### Official Review · Reviewer_JFFa · 2025-10-30

**Soundness:** 3
**Presentation:** 4
**Contribution:** 3
**Rating:** 6
**Confidence:** 2

**Summary:**

This paper studies inference-time scaling under the Pass@k framework, where the algorithm may generate up to $N$ responses but submit at most $k \leq N$ of them. In this paper, the evaluation of algorithms focus on (1) the scaling of regret in $k$ and (ii) a notion called "scaling-monotonic", i.e. whether the algorithm degrades as $N$ grows (when the reward model is accurate.)

The authors show the limitations of two common baselines: (Weighted) majority voting suffers a constant regret lower bound even when the reward model is perfect; also, Best-of-N (BoN) admits a lower bound violating scaling-monotonicity. As a fix, they propose an algorithm called *Best-of-Majority (BoM)*, which can be viewed as a "combination" of both: BoM (1) first filters to responses whose empirical frequencies are high (a majority-style step), and then (2) selects the top-$k$ by reward (a BoN-style step). Then they show that BoM achieves an upper bound matching the minimax lower bound they show for the setting. Crucially, BoM scales with $k$ (in addition to an optimal estimation error of the reward) and satisfies the scaling-monotonicity.

Empirically, they compare BoM against majority voting and BoN. The experimental results show that BoM generally outperforms or matches the baselines -- the plots versus $k$ show BoM consistently better on MATH-500 and competitive on GSM8K/AIME24, especially for small $k$.

**Strengths:**

- This paper is well-written, with a clear structure and flow of logic. It isolates the scaling-monotonicity (which is defined formally) as a target property and then set it as an explicitly goal of algorithms. By showing how baseline approaches fail this goal, and the present a lower bound and a simple algorithm that matches it, the line of work feels convincing and clean.

- I think the story that motivates the construction of the BoM makes good sense: Indeed, an ideal algorithm should balance the considerations on both uncertainty and the reward. For the former, BoM uses the empirical frequencies of generated responses to approximate the probability, and for the later, BoM then query the reward model on the surviving candidates.

**Weaknesses:**

(I should probably stress that the topic of this paper is not my area. Thus I may not have the expertise to fully assess e.g. the novelty and relevance of this work.)

- The uniqueness-of-optimum assumption feels restrictive. One can easily think of many tasks that admit multiple equally good answers (or near-ties). It seems that several bounds and the neat identification of $\mathcal{C}^*$ relies on the uniqueness assumption.

- It seems that BoM needs a coverage threshold that depends on $\mathcal{C}^*$, which can be unknown and may be badly misestimate. This paper does not offer, e.g. a data-driven estimator or a quantification of how misspecification affects regret or monotonicity.

- The empirical scope feels a bit narrow: all datasets are math/verifier-style. Given this, I'm not sure whether the empirical results are representative on a broader set of tasks.

**Questions:**

Could you respond to the points listed in the Weakness section? (e.g. discuss how sensitive the guarantees are when there are several maximizers or when rewards are flat within a small margin, etc.)

---

> ### Author Response · Authors · 2025-11-21
>
> Many thanks for your positive review!
>
> ---
>
> **Q1**: The uniqueness-of-optimum assumption feels restrictive. One can easily think of many tasks that admit multiple equally good answers (or near-ties). It seems that several bounds and the neat identification of relies on the uniqueness assumption.
>
> **A1**: The uniqueness-of-optimum assumption indeed plays an important role in our analysis. When multiple maximizers exist, the coverage coefficient $C^\*$ becomes dependent on the particular choice of the optimal policy $\pi^\*$. For example, if we have two optimal actions $A$ and $B$, then any policy supported on the set $\{A,B\}$ is optimal. However, when $\pi_{\text{ref}}$ is not uniform, the definition of the coverage $C^\* = \sup_{x}\frac{\pi^*(x)}{\pi_{\text{ref}(x)}}$ will rely on the selection of $\pi^\*$. This dependence on an arbitrary choice is undesirable, as it introduces unnecessary ambiguity into the theoretical quantities.
>
> On the other hand, many tasks with a unique correct answer, particularly mathematical problem solving and reasoning tasks, are already widely regarded as central benchmarks for test-time scaling. Therefore, we believe that focusing on the unique-solution case, which aligns with the dominant setting in prior work, is reasonable. We will leave the study of the problem with multiple optima as an interesting future direction.
>
> ---
>
> **Q2**: It seems that BoM needs a coverage threshold that depends on $C^*$, which can be unknown and may be badly misestimate. This paper does not offer, e.g. a data-driven estimator or a quantification of how misspecification affects regret or monotonicity.
>
> **A2**: We introduce the parameter $\alpha$ to make our algorithm scaling monotonic. In comparison, for Pass@1, the proposed algorithm **InferenceTimePessimism** in [1], which guarantees that the algorithm is scaling-monotonic, also introduces an additional parameter $\beta$ whose optimal value depends on problem-specific quantities (Theorem 4.1 in [1]). Therefore, this design choice is not unique to our method. We believe this is the price we need to pay for getting a scaling monotonic algorithm.
>
> Empirically, we do not directly estimate $C^*$. Instead, we treat the parameter $\alpha$ as a tunable parameter. We can test the performance of the algorithm on an additional holdout dataset, together with the grid search method to find the optimal selection of $\alpha$.
>
>
> ---
>
> **Q3**: The empirical scope feels a bit narrow: all datasets are math/verifier-style. Given this, I'm not sure whether the empirical results are representative on a broader set of tasks.
>
> **A3**: Mathematical tasks are widely used to assess the reasoning capabilities of LLMs, and numerous prior works have proposed new methods within this setting. A key property of mathematical reasoning problems is that they allow for diverse reasoning paths while having a uniquely verifiable final answer. This gives the problem a distinctive property and makes it worthy of special study. For instance, majority voting is well defined for mathematical problems, while in many other settings it may not work. Thus, we believe studying the math/verifier-style problem is already important enough.
>
> ---
>
> [1] Is Best-of-N the Best of Them? Coverage, Scaling, and Optimality in Inference-Time Alignment, Huang et al. 2025

---

### Official Review · Reviewer_ckrW · 2025-11-04

**Soundness:** 2
**Presentation:** 2
**Contribution:** 2
**Rating:** 2
**Confidence:** 5

**Summary:**

The paper studies Pass@k inference and proves that majority voting and Best-of-N are not scaling-monotonic, then introduces Best-of-Majority (BoM).

**Strengths:**

An optimal algorithm for inference-time algorithm

**Weaknesses:**

**Weaknesses**

1. **Relation to [1].** Aminian et al. [1] also derive an **upper bound** on BoN regret. Please state their result precisely, compare assumptions and regimes (dependence on (N), and reward-model error), and explain how your bounds differ or complement theirs.

2. **Comparison to InfAlign [2].** Please include an empirical and conceptual comparison to **InfAlign** [2], clearly outlining the methodological differences and where each approach is expected to excel or fail.

3. **Soft-BoN vs. BoN vs. yours.** As noted in [1], BoN can over-optimize; **Soft-BoN** is proposed as a mitigation. How does **Soft-BoN** perform relative to your method—both theoretically (regret guarantees, assumptions) and empirically?

4. **Asymptotics in Theorem 4.2.** Using Theorem 4.2, in the limit $(N\\to\\infty)$ the regret for a non-zero error appears to approach a **constant independent of (N)**. Please elaborate on the mechanism, the constant’s dependence on problem/reward parameters, and its practical implications.

5. **Reward calibration.** Is the **reward function calibrated**—either in the formulation (assumptions/guarantees) or in experiments (procedure, metrics)?

6. **“Mathematical equivalence.”** When you write “select (k) answers (up to **mathematical equivalence**) with highest frequency,” what exactly constitutes equivalence (e.g., algebraic identity, numerical equality within tolerance, format-invariant parsing)? Please define the criterion and how it’s implemented in experiments.

**References**

[1] Gholamali Aminian, Idan Shenfeld, Amir R. Asadi, Ahmad Beirami, Youssef Mroueh. *Best-of-n through the smoothing lens: KL divergence and regret analysis.* arXiv:2507.05913, 2025.

[2] Ananth Balashankar et al. *InfAlign: Inference-aware language model alignment.* ICML, 2025.

**Questions:**

please see weaknesses

---

> ### Author Response · Authors · 2025-11-21
>
> We appreciate your review and valuable suggestions!
>
> ---
>
> **Q1**: [1] also derive an upper bound on BoN regret. Please state their result precisely, compare assumptions and regimes (dependence on (N), and reward-model error), and explain how your bounds differ or complement theirs.
>
> **A1**: We will compare the results from various perspectives.
>
> **Assumptions**: [1] made 3 assumptions,
> 1. Bounded reward function;
> 2. Reward estimation error $\epsilon_{\beta,r}(x) := \frac{1}{\beta} \log(\mathbb{E}\_{\pi_{\text{ref}}}[\exp(\beta(r^*(x,y)-\hat r(x,y))^2)])$. This aligns with our Assumption 3.1 when $\beta = 0$;
> 3. Maximal reward can be achieved for the estimated reward $\hat r$. In comparison, we make Assumption 3.2 regarding the maximal reward: (1) The maximal reward is achieved by a unique response $y^\*$ such that $r^\*(x, y^\*)=1$, and (2) the error of the estimated reward $\hat r$ at $y^*$ is $\epsilon_{\mathrm{opt}}$.
>
> **Results**: [1] proved that the regret of BoN satisfies
> $$
> \text{Regret}(x)\le\sqrt{\epsilon_{\infty,r}(x)}\big(\sqrt{C_{\infty,\hat r, \text{ref}}(x)}+ \sqrt{C_{\infty,r^\*, \text{ref}}(x)}\big) + c \sqrt{\log\Big(1+\frac{C_{\infty,\hat r, \text{ref}}(x)-1}{N}\Big)},$$
> where it considers two coverage coefficients dependent on $\hat r$ and $r^*$, respectively. Moreover, the reward estimation error is characterized at $\beta=\infty$, which corresponds to the supreme norm, instead of the 2-norm. When $N\rightarrow\infty$, the convergence rate is $\sqrt{1/N}$.
>
> **Comparison with our results**:
> One central distinction lies in our different focuses. Our work focuses on analyzing algorithmic performance within the Pass@$k$ framework, and therefore our theoretical results explicitly account for the ability to submit $k$ different answers, which is not studied in [1]. In particular, for the theory of BoN, our Theorem 4.2 establishes a **lower bound** that depends explicitly on $𝑘$, which serves as a parallel result to [1].
>
> As for upper bound of BoM, we prove that it has an exponential decay dependence of $N$ (See Line 858), together with a term of $\sqrt{C\epsilon^2/k}$, which directly has $k$ dependence. Although the definition of reward error differs from that used in [1], thus the results can not be directly compared, we have shown a faster convergence rate of $N$ than [1].
>
> ---
>
> **Q2**: Please include an empirical and conceptual comparison to InfAlign [2], clearly outlining the methodological differences and where each approach is expected to excel or fail.
>
> **A2**: The goal of our paper is to study test-time algorithms, which operate purely at inference and do not require modifying model parameters. In contrast, the paper mentioned by the reviewer [2], lies in the domain of alignment, rather than test-time algorithm design. The central question addressed in [2] is: "Can we better align a language model to be served with a known inference-time procedure?" Their method therefore focuses on designing a new reward function, given the knowledge of the inference-time procedure. Then the model parameters are updated with the new reward function and PPO. In contrast, our goal is to design a new inference-time algorithm. In our revision, we have added discussion of [2] in the related work that examines how inference-time procedures interact with training, clarifying the conceptual distinction between alignment approaches and the test-time algorithmic focus of our paper.
>
> ---
>
> **Q3**: As noted in [1], BoN can over-optimize; Soft-BoN is proposed as a mitigation. How does Soft-BoN perform relative to your method—both theoretically (regret guarantees, assumptions) and empirically?
>
> **A3**: We have discussed the difference of assumptions in **A1**. For SBoN, the regret satisfies
> $$
> \sqrt{\epsilon_{\beta,r}(x)}\big(\sqrt{C_{\infty,\hat r, \text{ref}}(x)}+ \sqrt{C_{\infty,r^\*, \text{ref}}(x)}\big) + c \sqrt{\log(1+\frac{C_{\infty,\hat r, \text{ref}}(x)-1}{N})} + \log(C_{\infty, r^\*,\text{ref}}(x))/\beta.
> $$
> Note that this upper bound holds only when $\beta \neq 0$, otherwise the last term will explode. Thus, the setting of [1] is never same as ours. Again, when $N\rightarrow\infty$, the convergence rate of SBoN is $\sqrt{1/N}$, compared with the exponential decay for BoM. Moreover, we need to highlight again our focus is on the Pass@$k$ setting, which has never been analyzed in [1]. We have added a section in the appendix (Appendix A) to give a comprehensive comparison of the theoretical results with [1].
>
> For empirical results, we have added the performance of SBoN in our revision, considering both variants: with and without reward calibration.

---

> ### Author Response · Authors · 2025-11-21
>
> **Q4**: Using Theorem 4.2, in the limit $N\rightarrow \infty$, the regret for a non-zero error appears to approach a constant independent of $N$. Please elaborate on the mechanism, the constant’s dependence on problem/reward parameters, and its practical implications.
>
> **A4**: This aligns with the overoptimization phenomenon of BoN that you have mentioned. As we discussed in Line 280-282, we construct the hard case by introducing multiple distinct “bad” answers that are assigned higher estimated rewards. Take $k=1$ as an example. Assume we have two responses $A$ and $B$ with $r^\*(A) > r^\*(B)$ and $\hat r(A) < \hat r(B)$. For this simple instance, BoN will select response $B$ as long as it is sampled. When $N \rightarrow \infty$, no matter how small $\pi_{\text{ref}}(B)$ is, the probability of BoN selecting the suboptimal response $B$, i.e., $P[B \text{ exists in N independent samples from } \pi_{\text{ref}}]$ will approach $1$. Thus, BoN will suffer from a constant regret. We have also revised the proof of Theorem 4.2 to clarify the origin of this constant.
>
> In practice, it implies that for BoN, blindly increasing the computation budget $N$ will not always help. A suitable selection of the value $N$ will lead to improved performance, which is reflected in our experiments.
>
> ---
>
> **Q5**: Is the reward function calibrated—either in the formulation (assumptions/guarantees) or in experiments (procedure, metrics)?
>
> **A5**: In [2], the reward calibration is defined using the win rate over the base model $\pi_{\text{ref}}$ which naturally yields a bounded reward function. We would like to make clear that reward calibration is replaced with assumptions restricting the reward functions in our theoretical analysis, and not required in the experiments.
>
> - **In our theoretical formulation**, we only assume that the reward function is bounded, together with the estimation error assumption (Assumption 3.1). We do not impose any specific transformation, such as reward calibration.
> - **In the experiments**, majority voting, BoN, as well as our BoM algorithm, rely solely on the **ranking** of rewards rather than their absolute values. Since reward calibration does not affect the ranking, we do not need to do that in the experiments. For SBoM, however, the numerical reward value directly influences performance, and in the experiments, we consider both cases: with and without the reward calibration method, where the reward calibration follows the method in [2].
>
> ---
>
> **Q6**: When you write “select (k) answers (up to mathematical equivalence) with highest frequency,” what exactly constitutes equivalence (e.g., algebraic identity, numerical equality within tolerance, format-invariant parsing)? Please define the criterion and how it’s implemented in experiments.
>
> **A6**: The implementation comes from the official [repository](https://github.com/QwenLM/Qwen2.5-Math) of Qwen, which provides a function that determines whether two LaTeX math expressions are equivalent. In particular, if both expressions are composed of digits, then they are compared by numerical equivalent. Otherwise the scripts compare the LaTeX expression by first converting them to `sympy` expression and then determine the equivalence via `sympy` built-in methods. We refer to the original [implementation](https://github.com/QwenLM/Qwen2.5-Math/blob/a45202bd16f1ec06f433442dc1152d0074773465/evaluation/grader.py#L73) for more details.
>
>
> ---
>
> [1] Gholamali Aminian, Idan Shenfeld, Amir R. Asadi, Ahmad Beirami, Youssef Mroueh. Best-of-n through the smoothing lens: KL divergence and regret analysis. arXiv:2507.05913, 2025.
>
> [2] Ananth Balashankar et al. InfAlign: Inference-aware language model alignment. ICML, 2025.

---

> > ### Comment · Reviewer_ckrW · 2025-11-26
> > **Thanks for response**
> >
> > Thank you for thoughtful response. I have the following comment and appreciate if authors can comment on that:
> >
> > You chose AceMath-7B-RM for the experiments, which, if I understand correctly, is a very strong reward model. I am curious about how the performance compares when using a weaker or smaller reward model.
> >
> > I revised my score based on current draft.

---

> > > ### Author Response · Authors · 2025-12-02
> > >
> > > Many thanks for the suggestion of investigating weaker reward models!
> > >
> > > We acknowledge that AceMath-7B-RM is a strong reward model; however, since all algorithms receive the rewards generated with the same reward model on the same set of responses, this does not weaken our conclusion. To further demonstrate that our findings are not tied to any particular reward model, we additionally evaluate the performance of various inference-time algorithms using InternLM2-reward [1]. Since this model is not specifically trained for mathematical reasoning, it serves as a weaker reward model for our mathematical tasks. The results of this supplementary experiment are reported in Appendix E and lead to the same overall conclusion.
> > >
> > > [1] Internlm2, technical report. Cai et al. 2024

---

### Official Review · Reviewer_h4Se · 2025-11-12

**Soundness:** 4
**Presentation:** 3
**Contribution:** 3
**Rating:** 4
**Confidence:** 3

**Summary:**

The paper considers the problem of how to use a given LLM, a proxy (potentially inaccurate) reward model, and $N$ responses to a prompt (sampled iid from the LLM), to select $k$ out of the $N$ responses. The performance metric for any selection scheme is the regret between the expected reward of the best response and the expected best reward of the $k$ selected items, according to a (unknown) ground-truth reward model, also termed the pass@k regret. Within this setting, the paper argues that existing inference strategies such as Best-of-$N$ and weighted majority vote do not perform satisfactorily in that they do not give $o(1)$ regret with $N \to \infty$ and the are not monotone, i.e., their performance can degrade with more samples $N$ and an accurate proxy reward model. The paper goes on to design a new algorithm (Best of Majority) that, in spirit, is a 'best of both worlds' solution, blending the strengths of both these strategies using an additional probability threshold hyperparameter $\alpha$. This algorithm is shown to achieve monotone scaling and $o(1)$ regret as $1/N$ and the proxy reward model error tend to $0$. The paper also exhibits a fundamental lower bound (in a worst-case sense with a hard instance) on the regret scaling as a function of the reward model approximation errors, the base policy coverage distribution and $k$.

**Strengths:**

- The problem formulated and studied in the paper is very timely and relevant. Inference efficiency and quality optimization at scale carries a high potential to make real-world impact in LLM pipelines.

- The paper's mathematical model of the inference problem is thorough, and captures several practical aspects, including the fact that reward model oracles used for cheaply judging quality of responses must be only approximately accurate, and that upto $k \geq 1$ responses could be proposed for a prompt in the pass@k performance metric.

- I appreciated the clear and precise technical analysis of the regret performance metric, using appropriate tools from probability and concentration methods. The results are presented with clearly stated hypotheses/assumptions and expose the underlying problem-dependent quantities clearly, e.g., $\epsilon_{RM}$, $C^*(x)$, etc.

- The paper's analysis is comprehensive, showing both algorithm-dependent and fundamental performance limits for the pass@k regret.

- The paper designs a new inference algorithm (best of majority aka BoM) to circumvent the weaknesses raised about the existing inference algorithms.

**Weaknesses:**

- While the negative results for existing inference algorithms such as BoN and weighted majority vote are illuminating, they are essentially of a worst-case flavor in the sense that a 'bad' instance is constructed in each case to render high $\Omega(1)$ regret. A concern is that this is too 'pathological'; instead, bringing out the dependence of an arbitrary problem instance and its parameters in the regret lower bound per algorithm could be more insightful and, of course, a stronger result that would imply, as special cases, the worst-case instances above.

- Perhaps the most significant weakness is that, while the new algorithm BoM ensures controlled regret, it also puts the burden on the deployment engineer of tuning the additional crucial hyperparameter $\alpha \in [0,1]$. The performance guarantee of BoM in the paper is only valid when $\alpha$ is set proportional to the probability of the base policy on an optimal answer, $\pi_{ref}(y^*|x)$, in addition to ensuring that $N$, the total number of generations, is above a threshold determined by the reward error $\epsilon_{RM}(x)$ **per prompt**. I wonder what practical modalities exist to be able to set these hyperparameters in this way. There is no adequate discussion of whether these hyperparameters can be estimated or learnt from data or trials.

- In a similar vein, the paper's experiments shed no additional light on how to set $\alpha$ and $N$ 'well' depending on the problem instance and reward model error. There is no discussion on what considerations went into the choice of $\alpha$ used in the experiments, as well as an ablation on how different values of $\alpha$ influence overall performance. These would be essential in judging the quality and robustness of the new algorithm.

- The dependence on $N$ (the total number of generations) is missing in Theorems 5.1 and 6.1. This is in contrast to the referenced work of Huang et al (2025) that clearly outlines how $N$ influences regret performance. Could the author(s) please discuss this explicitly?

**Questions:**

- Please see the 'Weaknesses' section for questions that are posed inline.

- (minor) Why bother defining two coverage coefficients $C^*(x)$ and $C^*_\infty(x)$ when you finally argue that they coincide?

---

> ### Author Response · Authors · 2025-11-21
>
> Many thanks for your feedback and valuable suggestions!
>
> **Q1**: The lower bounds for BoN and weighted majority voting are in a worst-case flavor in the sense that a 'bad' instance is constructed in each case to render high regret.
>
> **A1**: Our lower bounds for BoN and weighted majority voting (WMV) generalize the result from [1] to the Pass@$k$ setting, which is why our negative results follow the same worse-case flavour.
>
> From a statistical perspective, our lower bound follows the classical framework of the minimax lower bound (see, e.g., Section 13, 15 in [2]). The idea is to evaluate any algorithm by analyzing its performance on a carefully constructed hard instance. This enables us to compare the lower bounds with the upper bound of our algorithm (Theorem 5.1), which also works for the worst-case instance. Therefore, the minimax lower bound remains meaningful.
>
> The characterizzation of how the performance of different algorithms depends on the specific parameters of each problem instance aligns with the notion of *instance-dependent lower bounds* (see, e.g., Section 16 in [2]). The challenge of deriving the instance-dependent lower bound in our theoretical framework is that each instance contains multiple interacting parameters, such as the value of $\pi_{\mathrm{ref}}$, $\hat r$, and others, while different algorithms may rely on different aspects of the instance. We appreciate your suggestion about proving an instance-dependent lower bound. However, due to the technical difficulty and the primary focus of our paper on designing a **minimax-optimal** algorithm for the Pass@$k$ setting, we will leave it as future work.---
>
> **Q2**: The dependence on $N$ is missing in Theorems 5.1 and 6.1.
>
> **A2**: For Theorem 5.1, we would like to correct the misunderstanding of the reviewer: The regret upper bound of BoM contains a term dependent on $N$, but when choosing $N=\tilde\Omega(C^*(x))$, the term is dominated by others so that the Theorem 5.1 can be presented in a simpler form.
>
> In detail, we prove that the upper bound contains a term that decays exponentially in $N$, i.e., $\text{Regret}(x)\le \epsilon_{\text{opt}}(x) + O(\sqrt{C^\*(x) \epsilon_{\text{RM}}^2(x)/k}) + O(C^\*(x)\exp(-N/(32C^*(x)))$ (Line 858) which stems from the analysis of the high-probability event. In our revision, we have illustrated the dependence of $N$ more explicitly in the main text (Lines 344-347).
>
> Theorem 6.1 is a general lower bound for **any** algorithm in the minimax sense. This lower bound holds even for algorithms that do not select responses from the sampled responses, for which the sampling budget $N$ does not exist. The closest related result in [1] is Proposition 2.3, which likewise does not include any dependence on $N$.
>
> ---
>
> **Q3**: The algorithm requires tuning the parameter $\alpha$, in addition, it needs to ensure that, the total number of generations, is above a threshold determined by the reward error **per prompt**.
>
> **A3**: First, a minor clarification: Since the upper bound in Theorem 5.1 decays exponentially with $N$ (as discussed in A2), we can replace $\epsilon(x)$ with any accuracy $\epsilon$ that we hope to achieve. Thus, the total number of generations do not need to depend on the reward error **per prompt**.
>
> Regarding the additional tunable parameter, $\alpha$, we introduce it to make our algorithm scaling monotonic. In comparison, for Pass@1, the proposed algorithm **InferenceTimePessimism** in [1], which guarantees that the algorithm is scaling-monotonic, also introduces an additional parameter $\beta$ whose optimal value depends on problem-specific quantities (Theorem 4.1 in [1]). Therefore, this design choice is not unique to our method. We believe this is the price we need to pay for getting a scaling monotonic algorithm.
>
> ---
>
> [1] Is Best-of-N the Best of Them? Coverage, Scaling, and Optimality in Inference-Time Alignment, Huang et al. 2025
>
> [2]  Bandit Algorithm, Lattimore, Tor and Szepesvari, Csaba, 2020

---

> ### Author Response · Authors · 2025-11-21
>
> **Q4**: In a similar vein, the paper's experiments shed no additional light on how to set $\alpha$ and $N$ 'well' depending on the problem instance and reward model error. There is no discussion on what considerations went into the choice of $\alpha$ used in the experiments, as well as an ablation on how different values of $\alpha$ influence overall performance.
>
> **A4**: Empirically, we can test the performance of the algorithm on an additional holdout dataset, together with the grid search method to find the optimal selection of $\alpha$. We add a section on the ablation study of the parameter $\alpha$ (Section 7.3). When $\alpha=0$, BoM will be reduced to BoN, meaning that as $\alpha$ approaches 0, BoM behaves similarly to BoN. We further observe that selecting a larger $\alpha$ can potentially improve the performance when $k$ is small, although this comes at the cost of deteriorated performance for larger $k$.
>
> ---
>
> **Q5**: (minor) Why bother defining two coverage coefficients $C^\*(x)$, $C^\*_\infty(x)$ when you finally argue that they coincide?
>
> **A5**: In the literature, both definitions of coverage coefficients have been studied. In our setting, these two definitions coincide, so distinguishing between them is outside the scope of our work. We state the definitions explicitly to prevent readers from being distracted by their possible differences.

---

### Author Response · Authors · 2025-12-02
**Message to New AC**

Dear new Area Chair,

Thanks you for taking over our submission! We would like to support your evaluation process in light of the challenges ICLR faced this year.

We greatly appreciate the suggestions of all the reviewers and we believe that we have addressed all their concerns. We summarize our rebuttal and the author-reviewer discussions as follows:

- Reviewer ckrW at first gave a negative rating, and was concerned about the comparison against soft best-of-N (SBoN) and some other works. After reviewing our rebuttal, they also asked whether our method remains effective when using weaker reward models. We have fully addressed these questions by adding experiments on SBoN, discussing reward calibration, and evaluating with an alternative reward function. The manuscript has been updated accordingly. Beyond their suggestions on comparisons and presentation, which we have now fully addressed, we did not identify any remaining critical concerns. They were satisfied with the detailed comparisons provided in our rebuttal and raised the rating from 2 to 4 before the rollback, before seeing the new reward-model results.
- Reviewer Df6F asked questions about how the regret bound scales with the sampling budget $N$ and the alphabet size $|\mathcal{Y}|$. We explained that the regret actually contains a term that decays exponentially in $N$, and this term is dominated by other terms under the condition of $N=\tilde\Omega(C^\ast)$ in Theorem 5.1. Furthermore, the alphabet size does not affect the regret. Reviewer Df6F was also concerned with the choice of hyperparameter $N$ without knowledge of $C^\ast$ or $\epsilon_{\mathrm{RM}}$. We argued that our algorithm does not require knowledge of $\epsilon_{\mathrm{RM}}$, and that the dependence of $N$ on $C^\ast$ is not unique to our work. They had raised the rating from 6 to 8 before the rollback.
- Reviewer h4Se is another reviewer who maintained a negative opinion. One of their concerns focused on the nature of the worst-case lower bound. We clarified that this likely stems from limited familiarity with statistics or bandit theory, as analyzing minimax lower bounds is a standard and well-established practice in these fields. We also discussed the inherent difficulty of developing instance-dependent lower bounds. Another concern was the need to tune an additional hyperparameter. We explained that this hyperparameter is essential for ensuring the monotonic scaling behavior of our algorithm. In fact, prior work of Pass@1 also requires tuning another parameter to guarantee scaling monotonicity. We believe that it's the price we have to pay. Empirically, the hyperparameter can be selected in a practical manner, for example, by evaluating performance on a small hold-out set and performing a simple grid search. We have also added an ablation study on $\alpha$ in our revised manuscript.
- Reviewer kb59 offered several suggestions regarding the intuition behind certain technical concepts, which we have incorporated into our revision. The most central point in our discussion concerns the intuition behind the scaling monotonic property. An algorithm with this property will not violate the upper bound when the number of samples increases. After reviewing our rebuttal, Reviewer kb59 also agreed with the theoretical importance of this property and admired the empirical success. Our remaining discussion centers on how scaling monotonicity should be interpreted in relation to empirical observations. We would like to argue that the problem instances in practice can be different from the hard instances in theory. The experiments can also be subject to randomness in sampling. These two factors can contribute unexpected experiment results, but scaling monotonicity is still a useful guideline when choosing $N$, as we have provided an upper bound even in the worst case.
- Reviewer JFFa was concerned with the assumption of unique optimal response and the focus on math datasets in the experiments. We argued that the mathematical datasets, where each question has a unique verifiable answer, is one of the most important tasks that requires the **reasoning capability** of LLMs. They also asked questions about the choice of $\alpha$ similar to Reviewer h4Se.

During the rebuttal process, we received valuable and positive feedback from Reviewers ckrW, Df6F, and h4Se. Although we did not receive further responses from Reviewers h4Se and JFFa, we believe we have fully addressed their concerns and incorporated their suggestions regarding presentation. We would be grateful if these considerations could be taken into account when forming the final recommendation.

---

### Meta-Review · Area_Chair_wRLc · 2026-01-05

**Summary:**

The reviewers agree the paper makes a strong and timely theoretical contribution to inference-time scaling for LLMs under the Pass@k framework. The problem is well motivated, and the analysis clearly shows limitations of common strategies like majority voting and Best-of-N. The proposed Best-of-Majority method is simple, intuitive, and well justified, with matching upper and lower bounds proving minimax optimality, which reviewers found convincing. Initial concerns about worst-case lower bounds, tuning, and focus on math datasets were addressed in the rebuttal with clarifications, extra experiments, ablations, and stronger positioning relative to prior work. Overall, the combination of clean theory, clear separation results, and solid empirical validation led to a positive consensus, and the paper is considered a good fit for ICLR.

**Reviewer Concerns:**

The rebuttal clarified scaling monotonicity, dependence on sampling budget, and tuning of the extra parameter, and explained the link to related work. Additional experiments with weaker reward models and ablations improved robustness. Remaining concerns about generality beyond verifier-style tasks and practical tuning are acknowledged as limitations, not blockers.

**Reviewer Scores:**

Positive reviewers kept their scores. Several borderline reviewers raised theirs after the rebuttal, and at least one skeptical reviewer moved to recommend acceptance. Overall, the discussion shifted clearly toward acceptance.

---

### Decision · Program_Chairs · 2026-01-26

Accept (Poster)